# Dual-comb optomechanical spectroscopy

Xinyi Ren[1,8], Jin Pan[1,8], Ming Yan [1,2,3] ✉, Jiteng Sheng [1,4] ✉, Cheng Yang[1], Qiankun Zhang[1], Hui Ma[1], Zhaoyang Wen[1], Kun Huang [1], Haibin Wu [1,4,5,6] ✉ & Heping Zeng [1,2,3,7] ✉

Optical cavities are essential for enhancing the sensitivity of molecular absorption spectroscopy, which finds widespread high-sensitivity gas sensing applications. However, the use of high-finesse cavities confines the wavelength range of operation and prevents broader applications. Here, we take a different approach to ultrasensitive molecular spectroscopy, namely dual-comb optomechanical spectroscopy (DCOS), by integrating the high-resolution multiplexing capabilities of dual-comb spectroscopy with cavity optomechanics through photoacoustic coupling. By exciting the molecules photoacoustically with dual-frequency combs and sensing the molecular-vibration-induced ultrasound waves with a cavity-coupled mechanical resonator, we measure high-resolution broadband ( > 2 THz) overtone spectra for acetylene gas and obtain a normalized noise equivalent absorption coefficient of $1.71 \times 10^{-11} \, \text{cm}^{-1} \cdot \text{W} \cdot \text{Hz}^{-1/2}$ with 30 GHz simultaneous spectral bandwidth. Importantly, the optomechanical resonator allows broadband dual-comb excitation. Our approach not only enriches the practical applications of the emerging cavity optomechanics technology but also offers intriguing possibilities for multi-species trace gas detection.

Highly-selective and ultrasensitive gas sensing, with widespread applications ranging from breath analysis[1] to environmental monitoring[2,3], constantly demands novel spectroscopic approaches. The emergence of optical combs, a coherent light source consisting of massive equidistant, ultra-sharp frequency lines, has enabled many revolutionary approaches to molecular spectroscopy[4,5]. Particularly, dual-comb spectroscopy (DCS), harnessing two combs of slightly different line spacings and a fast, single-pixel detector, enables multiplexed spectral acquisition without the use of moving parts[6]. As a result, it allows the simultaneous detection of non-neighboring molecular characteristic absorption lines with unprecedented spectral resolution, bandwidth, precision, and speed. Despite being challenging, integrating these features with ultrahigh sensitivity has proven to be essential for the tasks like selective multispecies detection[3], reliable analysis of complex mixtures[1], and real-time monitoring of trace gases[2].

Ultrasensitive DCS has been demonstrated using hollow-core fibers[7], multi-pass cells[8], and optical resonance cavities[9–11]. Among these demonstrations, cavity-enhanced schemes, with unmatched pathlength enhancement, yield the lowest detection limits − possibly down to the parts-per-trillion (ppt) level[10]. The resonance cavities, however, restrict the wavelength range of operation due to the technical difficulty of fabricating broadband high-reflection mirrors. Also, cavity-enhanced DCS needs extra efforts for comb-cavity coupling[9] and intracavity dispersion control[10,11]. Meticulous electronic controls and cavity designs[10,11] may alleviate these difficulties at the cost of increased system complexity and limited applicability. More generally, the above systems with long light-molecule-interaction lengths are bulky and may suffer from large sample volumes and low gas exchange rates, preventing their applications for real-time, in-situ gas monitoring.

[1]State Key Laboratory of Precision Spectroscopy, East China Normal University, Shanghai 200062, China. [2]Chongqing Key Laboratory of Precision Optics, Chongqing Institute of East China Normal University, Chongqing 401120, China. [3]Chongqing Institute for Brain and Intelligence, Guangyang Bay Laboratory, Chongqing 400064, China. [4]Collaborative Innovation Center of Extreme Optics, Shanxi University, Taiyuan 030006, China. [5]Shanghai Research Center for Quantum Sciences, Shanghai 201315, China. [6]Shanghai Branch, Hefei National Laboratory, Shanghai 201315, China. [7]Jinan Institute of Quantum Technology, Jinan, Shandong 250101, China. [8]These authors contributed equally: Xinyi Ren, Jin Pan. ✉e-mail: myan@lps.ecnu.edu.cn; jtsheng@lps.ecnu.edu.cn; hbwu@phy.ecnu.edu.cn; hpzeng@phy.ecnu.edu.cn

Alternatively, one can achieve ultrasensitive gas sensing via enhanced photoacoustic spectroscopy (PAS), such as quartz-enhanced PAS (QEPAS)[12] and cantilever-enhanced PAS (CEPAS)[13,14]. In contrast to optical detection, these techniques work at any molecular absorption wavelength and offer extreme sensitivity, without backgrounds, for a small gas volume. Their detection limits have reached the ppt-level or below with the normalized noise equivalent absorption (NNEA) coefficient down to $10^{-12}$ cm$^{-1}$·W·Hz$^{-1/2}$, but primarily due to the combination of optical resonance cavities[14,15] and for one spectral element at a time, which has compromised the selectivity and reliability. Also, the narrow acoustic bandwidths of the cavity-enhanced photoacoustic (PA) systems (e.g. 1 Hz in ref. 15) limit their overall performances (such as acquisition speed and spectral width). Recently, comb-enabled multiplexed or broadband PAS[16–20] and photothermal spectroscopy[21] have been explored yet with sensitivities limited to sub-ppm (parts per million) levels or above, barely sufficient for trace gas detection. Hence, a novel strategy that improves the sensitivity within a wide acoustic bandwidth for real-time multiplexed PA sensing (that potentially works at any wavelength) is highly demanded.

Recently, cavity optomechanical sensors have attracted a great deal of attention and have been recognized as a promising type of ultrasensitive sensors, benefiting from the significant sensitivity enhancement by both high-quality mechanical and optical resonators. Cavity optomechanical sensors have been widely employed in a variety of applications, from gravitational waves[22] and dark matter detection[23] to displacement[24–26], acceleration[27], mass[28,29], and acoustic sensing[30,31]. Despite tremendous advancement, no experiments aimed at enhancing broadband molecular spectroscopy with cavity optomechanics have been reported.

Here, we demonstrate ultrasensitive multiplexed spectroscopy by combing DCS and cavity optomechanics. Under the dual-comb excitation, a multiplexed molecular absorption spectrum is downconverted to a heterodyne ultrasonic signal and then transferred to the vibration of a mechanical resonator. The membrane-in-the-middle (MIM) cavity optomechanical system[26] detects the vibration of a mechanical resonator in real time with a high displacement sensitivity. Such a dual-comb optomechanical spectroscopy (or DCOS) is promising for a wide scope of gas sensing applications due to its superior sensitivity and the advantageous spectral bandwidth, resolution, and acquisition speed.

## Results

### Basic principle

Figure 1 illustrates the principle of DCOS, which involves two main parts: (1) the ultrasonic generation utilizing dual combs as the excitation light and (2) the ultrasonic detection based on a MIM optomechanical system. The upper panel in Fig. 1a presents the typical dual combs of different line spacings. The two spatially-overlapped coherent combs function as a synthetic light source consisting of different components. The optical frequency (OF) of the $n$th component can be described as

$$f_{\text{OF}(n)} = \frac{1}{2}\left(f_n^{(1)} + f_n^{(2)}\right),\tag{1}$$

where $f_n^{(i)} = f_0^{(i)} + n \cdot f_r^{(i)}$ is the frequency of the $n$th line of each comb (distinguished as $i = 1, 2$) and $f_0$ denotes the offset frequency of a comb and $f_r$ the line spacing. The component is intensity modulated at a distinguished radio frequency (RF), i.e., the beat frequency of the $n$th paired comb lines, as

$$f_{\text{RF}(n)} = \left|f_n^{(1)} - f_n^{(2)}\right| = \triangle f_0 + n \cdot \triangle f_r,\tag{2}$$

where $\triangle f_0 = |f_0^{(1)} - f_0^{(2)}|$ and $\triangle f_r = |f_r^{(1)} - f_r^{(2)}|$. Suppose $f_{\text{OF}(n)}$ matches a molecular ro-vibrational transition and $\triangle f_0$ and $\triangle f_r$ are considerably small with respect to $f_{\text{OF}(n)}$; in that case, the $n$th component will be absorbed by the molecules and yield a PA wave at $f_{\text{RF}(n)}$ (due to the intensity modulation). Consequently, the interaction between many of

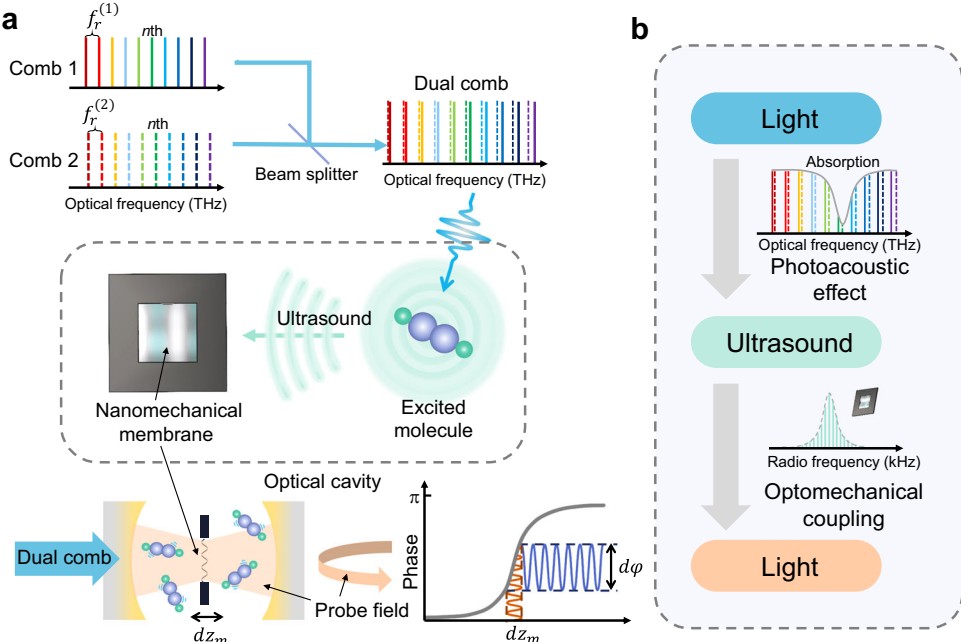

**Fig. 1 | Dual-comb optomechanical spectroscopy (DCOS). a** In DCOS, two combs excite the ro-vibrational transitions of gas molecules. The excited molecules release a portion of energy in the form of heat. The periodic heating, due to the beating between the paired dual-comb lines (within the absorption line profiles), yields ultrasonic waves which cause the displacement ($dz_m$) of a nanomechanical membrane placed in the middle of a high-finesse (finesse: $\mathscr{F}$) optical cavity. Meanwhile, a probe field (wavelength:$\lambda$) is coupled into the cavity from the opposite direction. The displacement induces a phase change ($d\varphi \propto \frac{\mathscr{F}}{\lambda}dz_m$)[36] of the probe field, which is magnified by the resonance between the field and the cavity and is detected interferometrically. **b** Illustration of the "light-sound-light" scheme. An ultrasound comb signal is produced by the dual-comb light via the photoacoustic effect and is then detected by the probe light through the optomechanical coupling.

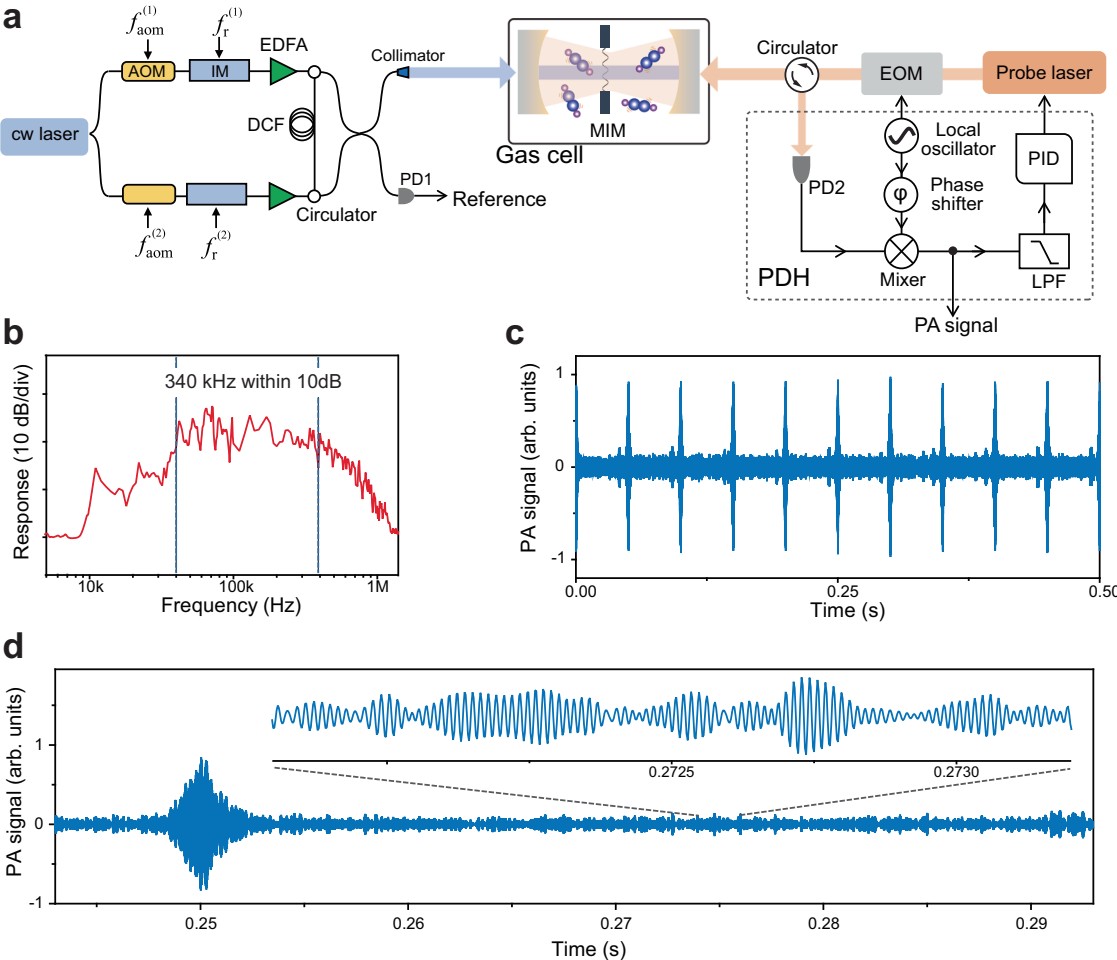

**Fig. 2 | Experimental setup and results. a** Experimental setup. A cw laser is used to seed two EO combs, each consisting of an acoustic-optical modulator (AOM), an intensity modulator (IM), and an erbium-doped fiber amplifier (EDFA). The two combs counter-propagate inside a dispersion compensation fiber (DCF) for flat-top spectral broadening. After that, they are combined with a 50:50 fiber coupler and then launched, through a fiber collimator, into the gas cell. A portion of the dual combs is detected by a photodetector (PD1) for the reference. For gas detection, a membrane-in-the-middle (MIM) cavity optomechanical system is utilized. This system consists of a nanomechanical membrane, a resonance cavity and a probe laser which is locked to the cavity via the Pound-Drever-Hall (PDH) technique (Supplementary Note 2). EOM electro-optic modulator, PID Proportional-Integral-Derivative, LPF electronic low-pass-filter. **b** The frequency response of the nanomechanical membrane. **c** Recorded interferometric PA signal for 10% $^{12}C_2H_2/N_2$. **d** The closer inspections of the interferometric PA signal in a measurement time window from 0.245 to 0.295 s.

the dual-comb lines and the molecules causes a series of superimposed PA waves[18–20], which vibrate the nanomechanical membrane immersed in the sample molecules, as shown in the middle panel in Fig. 1a. The displacement ($dz_m$) of the membrane is monitored at an ultrahigh sensitivity by recording the phase variation ($d\varphi$) of the reflected probe field from the MIM system (the lower panel of Fig. 1a). It is worth emphasizing that, unlike cavity-enhanced DCS, the optical cavity in DCOS is solely resonant with the single-mode probe field and therefore does not constrain the dual-comb excitation bandwidth.

As illustrated in Fig. 1b, such a "light-sound-light" process shows the developed scheme with incorporated advantages of different techniques: (1) the dual combs enable the rapid, high-resolution multiplexed spectroscopic interrogation, (2) the PA method provides the background-free measurement, and (3) cavity optomechanics makes the ultrasensitive detection possible by utilizing both mechanical and optical resonant enhancements.

### Experimental setup

Figure 2a depicts the experimental setup, including a dual-comb source and a MIM system for ultrasonic wave detection. In this proof-of-concept demonstration, two electro-optic (EO) combs are employed for several reasons. First, the two EO combs have excellent mutual coherence (exceeding 100 s without complicated phase control), as they share the same continuous-wave (cw) seed laser whose frequency ($f_{cw}$) is tunable from 192.17 to 197.23 THz (or 1520 to 1560 nm). Second, the combs have remarkable frequency agility. The comb center frequency, the line spacings, $f_r^{(i)}$, and the offset frequency difference, $\Delta f_0$, can be set arbitrarily and adjusted rapidly (Methods). Spectral broadening in a nonlinear fiber extends the number of comb lines to a maximum of 2400 lines per comb (for $f_r$ = 100 MHz), with a flat-top spectrum spanning 240 GHz or 2 nm within −10 dB (Supplementary Fig. 1). Besides, the EO combs, without mode-locking cavities, exhibit great flexibility, robustness and simplicity, benefiting practical applications. Since similar sources have been well documented in literature[7,19–21], we leave the details in Supplementary Note 1.

For ultrasensitive PA detection, a MIM system, consisting of two plano-concave mirrors resonant for the probe laser and a flexible stoichiometric silicon nitride nanomechanical membrane in the middle (Supplementary Fig. 2), is placed inside the cell (see Supplementary Note 2). The membrane has a thickness of 50 nm and a 2 × 2 mm² size (Supplementary Fig. 3). The mechanical Q factor of membrane is ~400 at atmospheric pressure near $f_M$ (= 68 kHz). Importantly, this

optomechanical sensor exhibits a broad "light-sound-light" response function (Methods), ranging from 10 kHz to 1.1 MHz with a −10 dB width of 340 kHz, as shown in Fig. 2b. We attribute this broadband response to the coupling between the membrane (at atmospheric pressure) and the substrate where the membrane sits on. The wide bandwidth of a sensor is crucial for fast and broadband measurements.

For spectroscopic sensing, we shine the dual-comb beam on the geometric center of the membrane. As shown in Fig. 2a, the dual-comb beam is coupled into the cavity through a cavity mirror at near normal incidence. We adjust the comb center frequency ($\sim f_{cw}$) and the parameters, $\triangle f_0$ and $f_r^{(i)}$, to match molecular absorptions and consequently produce ultrasonic waves at $f_{RF(n)}$, which are resonant with the membrane. A single-mode laser emits the probe light ($\sim 30$ μW at 1064 nm) that counter-propagates with the dual combs (Fig. 1a). The frequency of probe laser is stabilized to the optical cavity (finesse $\sim 12,000$) of the MIM system via the Pound-Drever-Hall (PDH) technique. The PDH error signal is used for measuring the PA signal. In addition, a small part of the dual-comb light is detected as the reference spectrum and acquired synchronously with the PA signal, for canceling the effect of the light intensity variations of the comb envelope.

Figure 2c exemplifies a part of the recorded PA signal (i.e., the interferograms). In this measurement, the cell is filled with 10% $^{12}C_2H_2$ diluted in $N_2$ (pressure: $10^5$ Pa, temperature: 295 K). The PA signal is originating from the interaction between the combs (centering at 195.378 THz with a total power of 1 mW) and the P(14) and P(15) transitions of $^{12}C_2H_2$ $\nu_1 + \nu_3$ band. We set the dual-comb parameters as $f_r^{(1)} = 400$ MHz, $\Delta f_r = 20$ Hz, and $\Delta f_0 = f_M = 68$ kHz so that all the dual-comb lines fall within the MIM bandwidth. The comb line spacing ($f_r^{(1)}$) suits the measurement of collision-broadened absorption lines (linewidths of several gigahertz) in the gas phase. In Fig. 2c, the interferogram refresh time ($1/\Delta f_r$) is 50 ms. An enlarged version of an interferogram, showing the superimposed PA waves, is given in Fig. 2d.

## Multiplexed spectral measurement

To obtain spectral information, we perform Fourier transform on a time-domain trace that lasts 2 s and contains 40 interferograms (similar to these in Fig. 2c). As a result, a comb-line-resolved spectrum is displayed in the RF domain in Fig. 3a. The resolved lines (insets in Fig. 3a), with Fourier-transform-limited linewidths, are evenly spaced by $\Delta f_r$ ($= 20$ Hz). Resolving these lines benefits high-resolution measurement, precise frequency calibration, and avoidance of instrumental lineshapes. However, in some cases (e.g. real-time gas monitoring[1–3]), short measurement times and high refresh rates are preferred over the above features (Supplementary Note 3). A spectrum resulting from a single interferogram of 50 ms is displayed in Fig. 3b. The refresh rate (set by $\Delta f_r$) is 20 Hz, and the measurement is performed in real time. Single-shot spectra measured at a refresh rate of $\Delta f_r = 2$ kHz, corresponding to a measurement time of 500 μs, are given in Supplementary Fig. 4. The spectral quality can be improved through either time-domain or frequency-domain averaging. For example, an improved spectrum resulting from the time-domain signal co-added in 1 s is also shown in Fig. 3b.

For spectroscopic validation, we compare experimental and simulation results in Fig. 3c. The simulation is performed using parameters from the HITRAN database. The spectral data (averaged in 1 s) are calibrated on the OF axis, using a conversion factor ($f_r^{(1)}/\triangle f_r = 2 \times 10^7$) and the correspondence between $f_{cw}$ and $\Delta f_0$ (see ref. 20). We also remove the spectral envelopes of the combs and the MIM sensor by using the PA spectrum divided by the convolution of the membrane response curve and the dual-comb spectral outline (Supplementary Fig. 1). In Fig. 3c, the normalized spectral data (orange

dots) agree with the simulation and their relative residuals are within 5%, with 1-σ standard deviation (SD) of 1.1 %.

The wavelength coverage of a single-shot spectrum is currently limited to the comb spectral width ($\sim 2$ nm or 240 GHz). This issue can be alleviated by stitching the normalized spectra obtained from several measurements where the dual-comb center frequency is tuned. Figure 3d shows a joint spectrum using data recorded from thirteen measurements. The joint spectrum spans 2.1 THz (from 194.20 to 196.30 THz), covering almost the entire P branch of $^{12}C_2H_2$ $\nu_1 + \nu_3$ band, and the spectral point spacing is 400 MHz (or ~3 pm). We also measure high-resolution spectra under gas pressure of $10^3$ Pa with $f_r^{(1)} = 100$ MHz (see Supplementary Fig. 5 and Supplementary Note 4). Note that, like any photoacoustic sensor[12–20], our system detects pressure waves, the strength of which decreases with reduced gas pressure. The above results manifest the high-resolution multiplexing capability of DCOS, which benefits accurate and highly-selective multi-gas analysis.

## Ultrasensitive detection

To investigate the sensitivity of DCOS, we use a standard gas mixture with 1 ppm $^{12}C_2H_2$ in $N_2$ at room temperature and one atmospheric pressure ($\sim 10^5$ Pa). We first confirm the detection enhancement by comparing the PA spectra recorded with and without the optomechanical cavity. For the latter, the reflection of an additional probe light field (which is not resonant with the optical cavity) from the membrane (reflectivity of 15%) is used to monitor the membrane displacement[32]. The results are plotted in the RF domain (Fig. 4a) and each spectrum is taken within 2 s. For this measurement, we tune the combs to 195.25 THz, targeting the $^{12}C_2H_2$ $\nu_1 + \nu_3$ band, P(17) transition. The optomechanically enhanced spectrum (blue) shows discernible spectral lines with a maximum SNR up to 90. The SNR is calculated by dividing the line peak by the SD of a noise background recorded with pure $N_2$. While, the spectrum (gray), without enhancement, shows barely noticeable lines with SNR close to 1. Thus, we obtain approximately two orders of magnitude enhancement for ultrasound detection with the help of the MIM system.

We plot enhanced spectra with different acquisition times in Fig. 4b. The spectra are normalized and displayed in the OF domain, together with a simulation curve for comparison. We evaluate the SNR of the detected PA signals versus acquisition times ($t$). In Fig. 4c, the SNR evolves as the square root of $t$ and reaches 900 at $t = 100$ s. The SNR is obtained for the peak of the absorption line. The noise-equivalent concentration (NEC) is 1.1 ppb (parts per billion) for measuring > 40 spectral elements simultaneously. Accordingly, we obtain an NNEA of $1.71 \times 10^{-11}$ cm$^{-1}\cdot$W$\cdot$Hz$^{-1/2}$ (Methods). The results manifest nearly two orders of magnitude sensitivity improvement compared to the existing comb-enabled PAS[16–20] and photothermal systems[21] (see Table 1). Also, the DCOS signals depend linearly on the gas concentrations and the excitation powers (Fig. 4d), favoring quantitative gas analysis.

To enable comparison with cw-based PAS, we determine the detection limit of our sensor by measuring Allan-Werle deviations[33–35] with a modulated cw laser (Methods). Figure 4e compares the results with the laser switched on (red) and off (blue). The former shows a minimum measurable concentration of 24 ppt (at 80 s), corresponding to an NNEA of $1.77 \times 10^{-11}$ cm$^{-1}\cdot$W$\cdot$Hz$^{-1/2}$, consistent with the dual-comb result ($1.71 \times 10^{-11}$ cm$^{-1}\cdot$W$\cdot$Hz$^{-1/2}$). The latter indicates the sensor's ultimate detection limit, i.e., 15 ppt for the integration time of 110 s (or $1.3 \times 10^{-11}$ cm$^{-1}\cdot$W$\cdot$Hz$^{-1/2}$). Note that one should distinguish this value (15 ppt for a single element) from the result of DCOS (1.1 ppb for 40 elements). Moreover, the two data sets in Fig. 4e are hardly distinguishable for $t < 80$ s, manifesting a negligible thermal effect. The electronic noise could be the dominant noise source (Supplementary Notes 5 and 6).

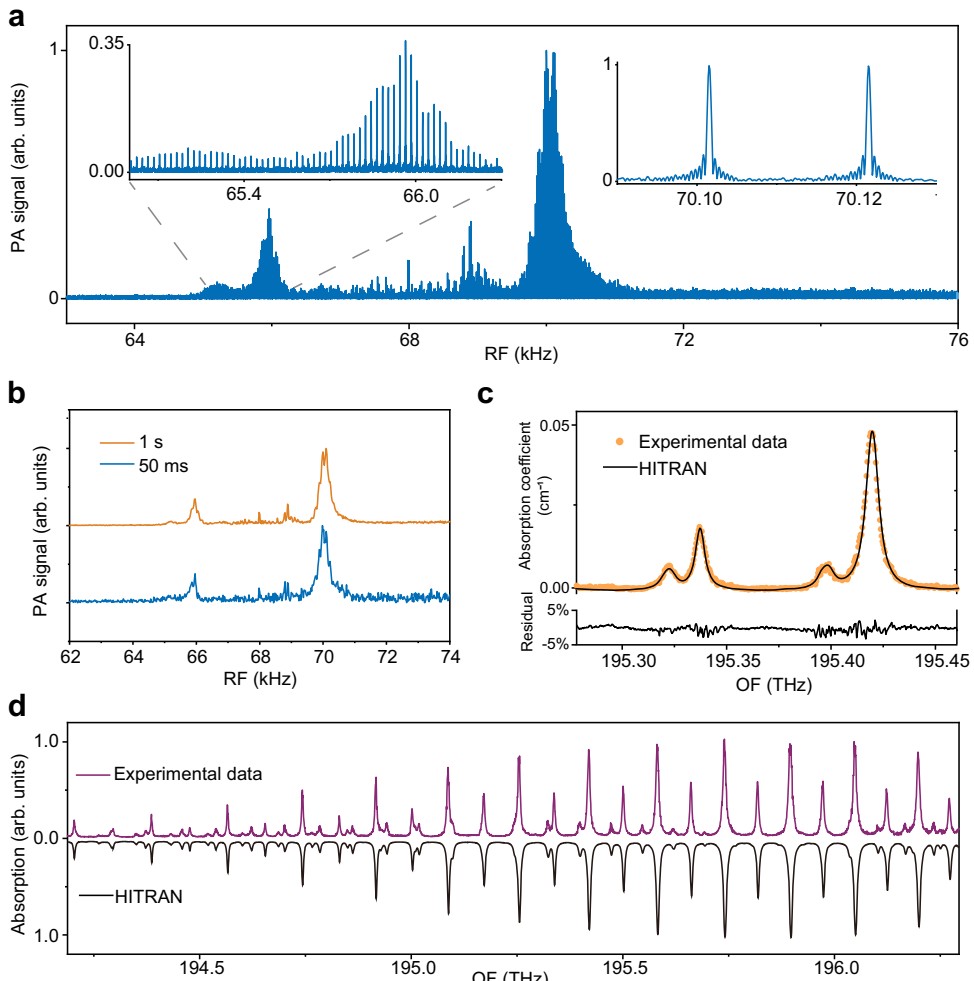

**Fig. 3 | Fourier-transformed DCOS. a** A Fourier-transformed PA spectrum with resolved comb lines. The acquisition time is 2 s. The side lobes of the resolved lines in the right inset are due to the apodized Fourier transformation. As such, these artifacts do not influence the measurements of molecular lines. Also, the spikes around 68–69 kHz are caused by the high-intensity lines at the center of the EO combs (Supplementary Fig. 1). **b** PA spectra obtained in a single shot measurement of 50 ms (blue) and with 20-fold averaging (orange) in 1 s. **c** Comparison between the experimental data (orange dots) and the HITRAN model (black line). **d** Broadband molecular spectra with a spectral range of > 2.1 THz. The point spacing for the experimental data (purple) is 400 MHz (or ~3 pm). The black line shows the HITRAN model for comparison. All the spectra in Fig. 3 are recorded with 10% $^{12}C_2H_2$ at atmospheric pressure.

## Discussion

DCOS has a similar essence to CEPAS, both of which measure tiny displacements with high sensitivity. Our experimental results show that DCOS has the advantage of combining the ultrahigh sensitivity and the wide ultra-acoustic bandwidth. Although the NNEAs for cavity-enhanced CEPAS[14] and QEPAS[15] have recently reached the level of $10^{-12}$ cm$^{-1}$·W·Hz$^{-1/2}$, the use of optical resonance cavities and the narrow detection bandwidths (e.g. 1 Hz in Ref. 15) prevent fast and broadband spectral measurements. Our results (NNEA, $1.71 \times 10^{-11}$ cm$^{-1}$·W·Hz$^{-1/2}$) are exceptionally good considering (1) the absence of the resonance cavities for enhancing molecular absorption and (2) the wide bandwidth for multiplexed PA detection. Detailed comparisons are given in Supplementary Table 1.

Furthermore, we compare our MIM system to a Michelson interferometer (shown in Supplementary Fig. 6) with the same displacement signal and find that the displacement sensitivity of our detection system is 1.2 fm/√Hz (Supplementary Fig. 7), which is within 10 dB away from the shot noise limit for the probe laser. Therefore, the sensitivity of the MIM system can be pushed to a tenfold increase when the system reaches the shot-noise limit (for the probe laser) by reducing the electronic noise from the detection circuit and optimizing the

system parameters, such as the optomechanical coupling strength and spatial overlap coefficient[36]. The sensitivity could be further improved with the help of phonon lasing[32] or even beyond the shot noise limit by using squeezing light[26]. Furthermore, accessing the mid-infrared region, where the absorption line strengths of fundamental transitions are typically one to two orders of magnitude stronger than that of the overtones in the near-infrared, will improve the sensitivity, possibly down to the ppt regime.

In contrast to conventional cavity-enhanced spectroscopy, our method bypasses the limitation on spectral bandwidth imposed by mirror coatings (Supplementary Figs. 8 and 9). The maximum spectral bandwidth ($\Delta v = \Delta B \cdot f_r / \Delta f_r$), determined by our acoustic bandwidth ($\Delta B = 340$ kHz within 10 dB), is 68 THz for $f_r = 400$ MHz and $\Delta f_r = 20$ Hz and could be enlarged by optimizing $f_r / \Delta f_r$. Currently, the EO combs limit the spectral bandwidth of our system (see Supplementary Note 7), which, however, can be overcome by using highly-coherent broadband combs[37,38].

Finally, we point out that, compared to the ultrasensitive methods relying on long light-molecule-interaction lengths, combing the dual combs and optomechanical sensors opens new opportunities for developing on-chip gas sensors with superior selectivity and sensitivity

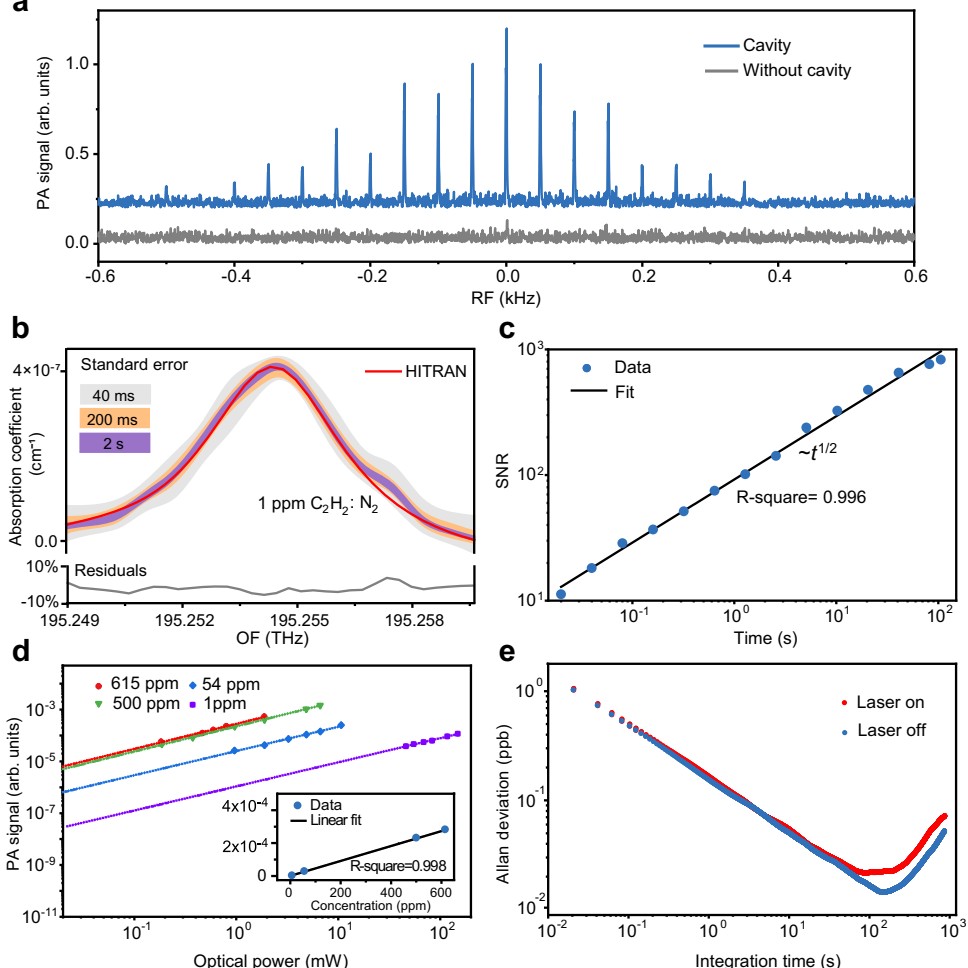

**Fig. 4 | Experimental results for ultrasensitive detection. a** Comb-line-resolved PA spectra measured with (blue) and without (grey) optomechanical enhancement, respectively, for 1 ppm $^{12}C_2H_2$/$N_2$. **b** PA measurements for the $^{12}C_2H_2$ $v_1 + v_3$ band P(17) line. A HITRAN model (red curve) is displayed for comparison. The shaded areas (light grey, orange, and purple) show the standard-error intervals for measurement times of 40 ms, 200 ms, and 2 s, respectively. The residuals between the 2-s data and the HITRAN simulation are plotted in grey. The systematic discrepancies around 195.258 THz are due to the additional electronic amplifier we used for amplifying the PA signal in the low-concentration measurement. **c** The SNR of the PA signal versus the acquisition time ($t$). The fitting (black) indicates that the SNR is proportional to $\sqrt{t}$ within 100 s. For A, B, and C, the dual-comb power is set to 150 mW. The number of comb lines is reduced to > 40 per comb for achieving a high power per comb line. Also, the line spacings are set to ~700 MHz with $\Delta f_r = 25$ Hz. **d** The excitation power dependences of the PA signal measured under different $C_2H_2$ concentrations (615, 500, 54 and 1 ppm, respectively). The inset shows the concentration dependence of the PA signal (with a fixed excitation power of 1 mW). The lines represent linear fitting. **e** Allan–Werle deviation analysis. The noise data of the MIM sensor over a period of 15 min are measured, with (red) and without (blue) switching on the excitation laser (200 mW), using a lock-in amplifier. The gas cell was filled with pure $N_2$.

since both technologies have shown the trend of micro-miniaturization[36,39].

In summary, we demonstrate the DCOS method, which offers ppb-level sensitivity without using resonance cavities (or equivalent devices) for enhancing molecular absorption and avoids the confinements on wavelengths, mirror coating, physical sizes, etc. The method potentially enables spectral measurements within the full spectral coverage allowed by optical combs (i.e., from ultraviolet to terahertz). The broad response bandwidth (340 kHz at −10 dB) is another merit of our system, which benefits multiplexed spectroscopic measurements in real time. Our method, unifying the two revolutionary techniques − DCS and cavity optomechanics, is promising for spectral measurement with broad spectral coverage, high resolution, short measurement times, and most importantly ultrahigh sensitivity. Integrating these features on a photoacoustic sensing platform brings new opportunities for selective and sensitive spectroscopic gas sensing and its applications, such as trace detection and multi-gas monitoring.

## Methods

### Dual EO combs

For generating the EO combs, a cw laser (frequency: $f_{cw}$; linewidth < 10 kHz) in the telecommunication C-band is split equally into two parts, each passing through an acoustic-optic modulator (driven frequency: $f_{aom}^{(i)}$ ~ 100 MHz) and subsequently an intensity modulator (modulation frequency: $f_r^{(i)}$). Consequently, two optical combs, each consisting of a set of comb lines at $f_n^{(i)} = f_0^{(i)} + n \cdot f_r^{(i)}$, where $f_0^{(i)} = f_{cw} + f_{aom}^{(i)}$ and $n = 0, \pm 1, \pm 2, \cdots$, are generated. The cw laser is constantly monitored by a wavemeter (WA-1650, Burleigh). The $f_{aom}^{(i)}$ and $f_r^{(i)}$ are individually set by four radio-frequency generators, which are disciplined to a hydrogen maser with a frequency stability of $10^{-13}$ in 1 s. The $f_r^{(i)}$ is tunable arbitrarily from 100 MHz to 1 GHz and the $f_{aom}^{(i)}$ is used for adjusting the dual-comb offset difference, $\Delta f_0 = \left| f_{aom}^{(1)} - f_{aom}^{(2)} \right|$. Each comb is amplified to maximally 300 mW with a home-made erbium-doped fiber amplifier, and then guided, through a fiber circulator, into a 200 m-long dispersion compensation fiber

**Table 1 | Comparison of comb-enabled PAS and photothermal systems**

| Ref. | Method | Sensing type | Gas | $\lambda$ μm | Simultaneous spectral width cm$^{-1}$ | Resolution cm$^{-1}$ | $P$ mW | $t$ s | NEC ppb | NNEA cm$^{-1}\cdot$W$\cdot$Hz$^{-1/2}$ |
|---|---|---|---|---|---|---|---|---|---|---|
| 16 | FTS | Cantilever-enhanced | $CH_4$ | ~3.3 | ~200 | 0.03 | 48 | 200 | 800 | $8 \times 10^{-10}$ |
| 17 | FTS | Cantilever-enhanced | $CH_4$ | ~3.3 | ~200 | 0.02 | 95 | / | 83 | / |
| 19 | Dual-comb | Microphone | $C_2H_2$ | 1.53 | ~1 | 0.03 | 20 | 1000 | 10000 | / |
| 20 | Dual-comb | Quartz-enhanced | $C_2H_2$ | 1.53 | 2 | 0.03 | 270 | 100 | 8.3 | $7 \times 10^{-10}$ |
| 21 | Dual-comb | Photothermal | $C_2H_2$ | 1.53 | ~2 | 0.017 | 15 | 1000 | 8700 | / |
| This work | Dual-comb | Optomechanically enhanced | $C_2H_2$ | 1.53 | 1 | 0.015 | 150 | 100 | 1.1 | $1.71 \times 10^{-11}$ |

*FTS* Fourier-transform spectroscopy, $\lambda$ wavelength, *P* Total excitation power, *t* Integration time, *NEC* Noise-equivalent concentration, *NNEA* Normalized noise-equivalent absorption coefficient.

(DCF, dispersion, −100 ps/nm·km; loss, 0.6 dB/km; nonlinear coefficient, < 10 W$^{-1}$km$^{-1}$) for spectral broadening.

### Data acquisition and processing
In our experiments, both the PA signal and reference are digitized with a 16-bit acquisition card (ATS9626, AlazarTech) at a sample rate of 500 kHz. The time-domain traces are then Fourier-transformed with 3-fold zero padding and triangular apodization.

### Resonance spectrum of membrane
For measuring the frequency response of the membrane (Fig. 2b), we employ an intensity-modulated cw laser fixed at 195.255 THz (targeting the $\upsilon_1 + \upsilon_3$ P (17) line of $^{12}C_2H_2$) for generating the ultrasound wave. We scan the modulation frequency across a wide RF range (the excitation power maintains 200 mW) and record the optomechanical signal. The measurement includes the influence of the collision-induced vibrational to translational relaxation, which differs for different gas species. In the case that the influence is significant, background calibration is needed. While, in many other cases, such as trace detection, the contents of targeted gas species (mainly mixed with air or $N_2$ at atmospheric pressure) are low, and the influence may be negligible. The result in Fig. 2b is for $C_2H_2/N_2$ gas mixture and the curve is highly reproducible in our experiments.

### Calculation of NNEA
For Fig. 4b, the minimum detectable absorption coefficient, $\alpha_{min}$, is $4.56 \times 10^{-10}$ cm$^{-1}$, calculated from the HITRAN database (http://www.hitran.com). Provided the excitation power per spectral line is $P_N = P/$ N = 3.75 mW (for $P = 150$ mW and N = 40), the normalized noise equivalent absorption (NNEA) coefficient, defined as $P_N\cdot\alpha_{min}\cdot\sqrt{t}$, is $1.71 \times 10^{-11}$ cm$^{-1}\cdot$W$\cdot$Hz$^{-1/2}$, where $t = 100$ s. From Fig. 4e, we obtain $\alpha_{min} = 6.2 \times 10^{-12}$ cm$^{-1}$ (laser off) and $9.9 \times 10^{-12}$ cm$^{-1}$ (laser on) for calculating the corresponding NNEAs.

### Allan-Werle deviation
For Allan-Werle deviation measurements[33,34], we use the intensity-modulated cw laser (the same one for seeding the EO combs) as the excitation beam and fill the cell with pure $N_2$ at atmospheric pressure ($\sim 10^5$ Pa) and room temperature (295 K). The intensity modulation frequency is fixed at $f_M$ (= 68 kHz). The output of MIM system is measured by a lock-in amplifier (HF2LI, Zurich Instruments). For the plots in Fig. 4e, the voltage signal is converted to the equivalent gas concentration, using a coefficient of 305.2 μV/ppm for $^{12}C_2H_2$, obtained from a calibration measurement[33–35] using the 200 mW cw laser.

### Data availability
The data used in this study are available in Zenodo under accession code DOI link. https://doi.org/10.5281/zenodo.8206975

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

## Acknowledgements

This research was supported by the Innovation Program for Quantum Science and Technology (2021ZD0302100, H.W.), National Key R&D Program of China (2022YFA1404202, J.S.), NSFC (12022411, M.Y.; 62035005, H.Z.; 11925401, H.W.; 12234008, H.W.; 12222404, J.S.; 11974115, J.S.), Shanghai Municipal Science and Technology Major Project (2019SHZDZX01, H.W.), Shanghai Pilot Program for Basic Research (TQ20220103, J.S.; TQ20220104, K.H.), Natural Science Foundation of Chongqing (CSTB2022NSCQ-JQX0016, M.Y.), Fundamental Research Funds for the Central Universities (M.Y., K.H.).

## Author contributions

M.Y., J.S., H.W., and H.Z. conceived the idea and designed the experiments. X.R. and J.P. conducted the experiment, analyzed the data, and drafted the manuscript. C.Y. and Q.Z. build the resonance cavity. H.M., Z.W., and K.H. build the comb source. M.Y., J.S., H.W., H.Z., and K.H. revised the manuscript. All authors provided comments and suggestions for improvements.

## Competing interests

The authors declare no competing interests.
