## [Peer Review File · Nature Communications]

Dual-comb optomechanical spectroscopyReviewer #1 (Remarks to the Author):

I have read through carefully the manuscript entitled "Dual-comb optomechanical spectroscopy" for Nature Communications. I conclude that this work is a unique extension of dual comb spectroscopy with the cavity optomechanics to molecular spectroscopy. The authors obtain high-resolution molecular spectra covering large wavenumber regions in short time. The sensitivity is also excellent as is seen from a well-presented comparison with other works. As far as I know, this is the first time this method has been applied to spectroscopy.

This manuscript is written very well.

This contribution is major step forward in experimental laser spectroscopy. I recommend that this contribution is accepted for publication.

Reviewer #2 (Remarks to the Author):

The authors present a method for the use of a cavity optomechanical sensor as the detection element of a dual comb spectrometer. The approach combines high sensitivity, flexibility, and spectral bandwidth. I believe the work is important and should be published in Nature Communications once a few small comments have been addressed.

1. I think it would be more useful if the abstract and introduction discussed the sensitivity of this approach and other approaches in terms of the normalized noise equivalent absorption rather than concentration (i.e. ppm/ppb) units. These type of concentration units are highly dependent on the selected gas and experimental conditions and make direct comparisons between methods difficult.

2. I would like to see the first table from the supplementary material moved to the main text. I think this kind of direct comparison between approaches is important.

3. I would expand and reword the sentence found on lines 53-55 (beginning "The resonance cavities"). There are a significant of number of concepts alluded to in this sentence and I think it would aid the reader if they were more explicitly described.

4. I think it would be useful if the manufacturer of the dispersion compensation fiber was given in either the main text or the supplemental.

Reviewer #3 (Remarks to the Author):

This manuscript describes the first demonstration of a dual-comb system combined with an optomechanical sensor based on a membrane placed in the middle of a cavity resonant for a probe laser. The system is used to detect C₂H₂ at 1.53 μ m. This a neat proof-of-concept demonstration and a new development within the field of DCS combined with photoacoustic sensing. However, the paper cannot be published in its current form, because the claim that this approach combines wide bandwidth and ultrahigh resolution (at fast acquisition time) is not supported by the shown data. In particular, the sensitivity quoted in the abstract is overestimated by almost 2 orders of magnitude.

Below I first write comments on the topics of bandwidth, sensitivity, and compactness of the system, then I list comments to the different sections of the paper.

BANDWIDTH

The simultaneous bandwidth shown in this work (2 nm/240 GHz) can be achieved without any problems with conventional cavity-enhanced comb-based techniques with the comb locked to the cavity. Even 40 nm (the entire tuning range of the comb used in this work) can be simultaneously transmitted by commonly available near-infrared mirrors. The total bandwidth of 2.1 THz

demonstrated here is obtained by stitching 13 spectra, which is hardly optimum or desired for real-time broadband sensing. It increases the measurement time by a factor of 13, not counting the dead time needed to tune the laser.

A discussion about the possible extension of the simultaneous bandwidth of this approach must be added, and the limitations should be discussed. The authors should at least calculate how large optical bandwidth one can fit into the bandwidth of the MIM system depending on the choice of f_{rep} and Δf_{rep} .

In this approach, the cavity needs to be resonant for the probe laser, but not for the comb. A discussion about the limitation to this approach is needed, considering mirror coatings or the choice of probe and comb laser wavelengths.

SENSITIVITY

The 1.1 ppb detection sensitivity at 100 s calculated from the SNR of the molecular signal is the limit that should be stated in the abstract. The authors instead state a 15 ppt detection limit, which I believe is calculated incorrectly and is not supported by the data. See comments under ULTRASENSITIVE DETECTION section below. For the 15 ppt detection limit, the authors characterized the noise in the system under different conditions than used for molecular detection, and it is not clear how they would justify the 100-fold improvement.

To achieve the 1.1 ppb detection limit, all comb power was channeled into very few spectral components (150 mW and >40 comb lines, spaced by 700 MHz). For a broader comb, the sensitivity will get worse, roughly by the amount of broadening, if the total power stays the same. Thus, this technique does not combine a wider bandwidth with the 1.1 ppb sensitivity.

The C₂H₂ lines detected in this work come from a combination band that is particularly strong in the near-infrared, with line strengths of the order of 10⁻²⁰ cm/molec. The 'typical' line strengths in the MIR are 10⁻¹⁹ to 10⁻²⁰ cm/molec, e.g. for methane at 3.3 μm, so the absorption sensitivity improvement will be at most one order of magnitude, not 2-3 as stated in the conclusions.

COMPACTNESS

Compactness is a subjective criterion, but if the authors want to claim that their system is particularly compact, they should discuss the physical size of the device or show a photograph. Considering the simplicity of the approach, the authors point out that the cavity does not need to be resonant for the comb; however, a cavity with PDH locking is needed for the MIM system.

OTHER ISSUES

The authors should discuss the limitation on the sample pressure; does the sensor operate only at atmospheric pressure? If yes, this will limit the applicability of this approach for multispecies detection because of spectral overlap of lines of different species.

It is not clear how the signal is extracted from the PDH error signal. In closed loop, i.e. under locked conditions, at frequencies within the locking bandwidth, the error signal should be zero. Is the bandwidth of the lock much lower than 70 kHz, the frequency at which the interferograms are detected? If not, how can the signal be observed there? See more comments under NOTE 4 below. English must be fixed at quite a few places.

BOTTOM LINE

The conclusions in this paper are not supported by the data. In particular, the authors claim

- Broad spectral coverage of 2 THz – but this is achieved by stitching of 13 spectra; the simultaneous bandwidth is 2 nm (240 GHz), and even smaller (40 x 700 MHz = 28 GHz) for the data used to evaluate the detection limit (i.e. to achieve the highest SNR).
- Millisecond measurement time – it is indeed demonstrated, but the 1.1 ppb sensitivity is achieved for 100 s acquisition time, and it will be 300 times worse at 1 ms.
- Detection limit of 15 ppt – this is wrong, the limit is 1.1 ppb at 100 s, see above and below.

INTRODUCTION

1. Line 48: Add references in the last sentence of this paragraph.
2. Line 49: The first sentence needs rephrasing, 'Ultrasensitive DCS has been investigated..'
3. Line 53: Provide references to works that achieve ppt sensitivities.
4. Line 54: The phrasing 'respecting mirror coating, comb-cavity coupling, dispersion' is not clear.
5. Line 68: 'limit overall performances' – please specify what performance parameter you are

referring to.

6. Line 73: demanding → demanded.

BASIC PRINCIPLE

7. Fig. 1a: It is not clear from this figure how the comb light is coupled to the cavity (from which direction). Please draw it.

8. Line 117: Specify what 'considerably small' means. Small with respect to what?

EXPERIMENTAL SETUP

9. Line 163: For clarity, add: mirrors 'resonant for the probe laser'.

10. Line 172: It is stated here that the comb beam is aimed at the geometric center of the membrane, but it should also be stated how it is coupled to the cavity (through the mirror? at an angle?).

11. Line 180: Reconsider the use of 'inhibiting', do you mean cancellation?

12. Line 184: Originated → originating.

13. Line 187: Locate → fall.

MULTIPLEXED SPECTRAL MEASUREMENT

14. Line 211: The PA spectrum should be baseline free, but here the authors discuss removing the baseline. This is contradictory and the origin of this baseline needs to be clarified.

15. Fig. 3b: Discuss the origin of the spikes next to the stronger line around 68-69 kHz.

16. Line 221, 228, and 316: 400 MHz is the sample point spacing, not the resolution.

ULTRASENSITIVE DETECTION

17. Line 240 and 248: It is not clear if the SNR is calculated on the peak of the absorption line or of a comb line. Clarify. It should be taken on the absorption line.

18. Fig. 4b: There are visible systematic discrepancies between the HITRAN model and the data. Plot the residuals and discuss the cause of the discrepancies.

19. Line 253: It is stated that there is an 'orders of magnitude sensitivity improvement' with respect to previous comb-based PA works. However, from table 1 in the supplementary it is clear that the improvement is less than 2 orders of magnitude. Rephrase this conclusion in the main text.

20. Line 257: Add reference to Werle's paper introducing the Allan-Werle method. It is Allan, not Allen.

21. Line 258: What does 'routinely' mean here?

22. Line 257 and on: It is not clear what CW laser is used here. The methods section, line 359, states that it is 'the same' laser. Same as what? Is that the seed laser of the EO combs? Specify. What is the amplitude of the applied intensity modulation compared to the amplitude of the interferogram signal? Allan-Werle deviation must be measured under the same conditions as the molecular signal from the sample. Here, it seems the authors characterized the noise in the system under different conditions. For example, they used a lock-in amplifier for the Allan-Werle plot, but not with the sample. How is the calibration coefficient (line 364) obtained? How is the voltage output of the lock-in amplifier compared to the interferogram amplitude? How can the authors explain the 15 ppt limit at 110 s, compared to 1.1 ppb at 100 s using the SNR method? What is the reason of this discrepancy of two orders of magnitude? I strongly suspect the 15 ppt detection limit is wrong.

DISCUSSION

23. Line 292 and on: The authors derive that the detection limit is 10 times above the shot noise. It is not entirely clear which of the detection limits they refer to (the 1.1 ppb or 15 ppt).

SUPPLEMENTARY

TABLE 1: The spectral bandwidth of 70 cm⁻¹ cannot be listed together with the NNEA of 1.7e-11 cm⁻¹ W Hz^{-1/2}, because the 70 cm⁻¹ bandwidth requires stitching of 13 spectra. The authors should either write the bandwidth of a single spectrum in the table, or recalculate the NNEA by increasing the measurement time by 13.

TABLE 2: In this table, the authors should list the demonstrated detection limit of 1.1 ppb, not 0.015 ppb (they actually write 0.027 ppb – not sure where this number comes from). According to this table, the current work does not supersede the sensitivity of CW based CE demonstrations.

NOTE 4:

It seems that the entire derivation is for an open loop error signal. It is not clear how the molecular signal is extracted in closed loop, under locked conditions. Its amplitude will depend on the gain in the feedback loop, this must be discussed.

- For Eq. (1), cite the original paper by Bjorklund instead of ref. S16.
- Above Eq. (4): 'overlap coefficient' – overlap of what?
- Above Eq. (5): State the LO frequency.
- Fig. 4 and below: Specify the pressure and power at which this dependence was derived. I see that these parameters are written in Eq. 11, but for clarity they should be stated together with the result of the calculation.

Manuscript NCOMMS-23-10200
“Dual-comb optomechanical spectroscopy”
Reply to Reviewers’ Comments:

We thank the Editor and the reviewers for carefully reviewing our manuscript. All the reviewers have provided positive comments on the novelty and advancement of our experimental results compared to the previous works in the field. The main concerns are the validity of the bandwidth and sensitivity claimed in our manuscript, raised by Reviewer 3, and a more precise comparison of sensitivity to other technologies suggested by Reviewer 2. In the revised manuscript, we have made corresponding modifications to the reviewers’ concerns and improved the readability thanks to the reviewers’ constructive suggestions. In the following, we address all the comments, clarifying the issues and improving the manuscript. The Reviewers’ comments are in black, and our replies are in blue (the modifications we made are in red).

Reply to Reviewer #1:

Reviewer #1 (Remarks to the Author):

I have read through carefully the manuscript entitled "Dual-comb optomechanical spectroscopy" for Nature Communications. I conclude that this work is a unique extension of dual comb spectroscopy with the cavity optomechanics to molecular spectroscopy. The authors obtain high-resolution molecular spectra covering large wavenumber regions in short time. The sensitivity is also excellent as is seen from a well-presented comparison with other works. As far as I know, this is the first time this method has been applied to spectroscopy.

This manuscript is written very well.

This contribution is major step forward in experimental laser spectroscopy. I recommend that this contribution is accepted for publication.

We thank the Reviewer for the positive assessment of our manuscript's originality and technical advancement.

Reply to Reviewer #2:

Reviewer #2 (Remarks to the Author):

The authors present a method for the use of a cavity optomechanical sensor as the detection element of a dual comb spectrometer. The approach combines high sensitivity, flexibility, and spectral bandwidth. I believe the work is important and should be published in Nature Communications once a few small comments have been addressed.

We thank the Reviewer for his report and constructive comments, which help improve our manuscript.

We are pleased that the Reviewer has a favorable opinion of our work, and we have revised the article following the comments.

1. I think it would be more useful if the abstract and introduction discussed the sensitivity of this approach and other approaches in terms of the normalized noise equivalent absorption rather than concentration (i.e. ppm/ppb) units. These type of concentration units are highly dependent on the selected gas and experimental conditions and make direct comparisons between methods difficult.

We followed the Reviewer's suggestions and modified the abstract and introduction using the normalized noise equivalent absorption.

We added the following sentence in abstract:

“... we measure high-resolution broadband (>2 THz) overtone spectra for acetylene gas and obtain a normalized noise equivalent absorption coefficient of $1.71 \times 10^{-11} \text{ cm}^{-1} \cdot \text{W} \cdot \text{Hz}^{-1/2}$.” (p. 2, line 10; original line 30)

We also added a sentence in the introductory part as:

“Their detection limits have reached the ppt-level or below with the normalized noise equivalent absorption (NNEA) coefficient down to $10^{-12} \text{ cm}^{-1} \cdot \text{W} \cdot \text{Hz}^{-1/2}$, but primarily due to the combination of optical resonance cavities^{14,15} and for one spectral element at a time, ...” (p. 4, line 8; original line 64)

2. I would like to see the first table from the supplementary material moved to the main text. I think this kind of direct comparison between approaches is important.

We thank the Reviewer for giving us such constructive advice. Following this advice, we moved Table 1 to the revised main text.

3. I would expand and reword the sentence found on lines 53-55 (beginning "The resonance cavities"). There are a significant of number of concepts alluded to in this sentence and I think it

would aid the reader if they were more explicitly described.

Following the Reviewer's suggestions, we modified the sentence as:

“The resonance cavities, however, restrict the wavelength range of operation due to the difficulty of broadband mirror coating. Also, cavity-enhanced DCS needs extra efforts for comb-cavity coupling⁹ and intracavity dispersion control^{10, 11}.” (p. 3, line 18; original lines 53)

4. I think it would be useful if the manufacturer of the dispersion compensation fiber was given in either the main text or the supplemental.

We added the manufacturer (YOFC; www.yofc.com) in Supplementary Note 1.

Reply to Reviewer #3:

Reviewer #3 (Remarks to the Author):

This manuscript describes the first demonstration of a dual-comb system combined with an optomechanical sensor based on a membrane placed in the middle of a cavity resonant for a probe laser. The system is used to detect C₂H₂ at 1.53 μm. This a neat proof-of-concept demonstration and a new development within the field of DCS combined with photoacoustic sensing. However, the paper cannot be published in its current form, because the claim that this approach combines wide bandwidth and ultrahigh resolution (at fast acquisition time) is not supported by the shown data. In particular, the sensitivity quoted in the abstract is overestimated by almost 2 orders of magnitude.

Below I first write comments on the topics of bandwidth, sensitivity, and compactness of the system, then I list comments to the different sections of the paper.

We thank the Reviewer for the detailed report and suggestions, which are precious for improving our manuscript.

The Reviewer pointed out that *the claim that this approach combines wide bandwidth and ultrahigh resolution (at fast acquisition time) is not supported by the shown data.*

We are grateful to the Reviewer for allowing us to clarify the bandwidth of our work. In our proof-of-concept demonstration, we showed simultaneous spectral acquisition of 240 GHz (or 2 nm) and a spectral coverage of 2.1 THz by stitching 13 spectra (with sample point spacing of 400 MHz or ~3 pm). In the revised version, we also added a Doppler-limited spectrum at 100-MHz point spacing.

Stitching several spectra to demonstrate the capability of broadband spectral detection is common in dual-comb spectroscopy [see *Phys Rev A* **84**, 062513 (2011); *Light: Science & Applications* **6**, e17076 (2017); *Nat. Commun.* **13**, 2181 (2022)]. For instance, the authors in ref. [21] (also *Nat. Commun.* **13**, 2181 (2022)) claimed that their system “provided opportunities for broadband gas sensing with super-fine resolution and high sensitivity” (a quote from ref. [21]) with a measured spectrum of 1 THz (by stitching 15 spectra at sample point spacing of 500 MHz).

More importantly, in our demonstration, we value the concept of combing dual-comb spectroscopy (DCS) and optomechanics which offers exciting opportunities for high-resolution broadband photoacoustic sensing because the optomechanical sensor had a wide acoustic bandwidth (340 kHz at -10dB). This concept is not bounded to the EO combs we used, which limited our simultaneous spectral width.

Particularly, we thank the Reviewer for advising us to discuss the maximum spectral bandwidth allowed by our optomechanical sensor (in the “*BANDWIDTH*” part). Indeed, our system permits simultaneous spectral detection over 68 THz (or ~500 nm) or even more if one replaces the EO combs with broadband mode-locked combs and adequately sets the dual-comb parameters (f_r and Δf_r). The details are given in the replies to “*BANDWIDTH*”.

Therefore, the wide bandwidth and ultrahigh resolution are supported by the principle of dual-comb optomechanical spectroscopy and our current optomechanical sensor. One can simply achieve ultra-wide bandwidth by replacing the EO combs with broadband mode-locked combs without affecting the detecting part, which is one of the advantages of our scheme, i.e., the comb excitation and ultrasonic detection are independent.

In order to clarify the claim of wide bandwidth and ultrahigh resolution, we have added the following discussion in the revised manuscript.

“In contrast to conventional cavity-enhanced spectroscopy, our method bypasses the limitation on spectral bandwidth imposed by mirror coatings (Supplementary Note 4). The maximum spectral bandwidth ($\Delta\nu = \Delta B \cdot f_r / \Delta f_r$), determined by our acoustic bandwidth ($\Delta B = 340$ kHz within 10 dB), is 68 THz for $f_r = 400$ MHz and $\Delta f_r = 20$ Hz and could be enlarged by optimizing $f_r / \Delta f_r$. Currently, the EO combs limit the spectral bandwidth of our system, which, however, can be overcome by using highly-coherent broadband combs^{36,37}.” (p. 17, line 13; original line 301)

The Reviewer also pointed out that *the sensitivity quoted in the abstract is overestimated by almost 2 orders of magnitude*.

We thank the Reviewer for allowing us to explain our sensitivity.

There are two concepts of sensitivity in our manuscript; one is the sensitivity measured with the combs (i.e., 1.1 ppb at 100 s for 40 spectral elements), and the other is the detection limit of the optomechanical sensor (i.e., 15 ppt at 110 s based on the intrinsic noise of the sensor). The latter was obtained from a standard Allan-Werle deviation measurement (Fig. 4 e, Laser off) when the excitation laser was switched off.

We highlighted the 15-ppt in the abstract because it represented the intrinsic characteristics of our optomechanical sensor and was essential for comparison with other photoacoustic sensors. In contrast, the 1.1-ppb was the sensitivity of our current dual-comb setup, which was affected by, e.g., the excitation comb power and the molecular line strengths.

We noticed that our original abstract could be misleading regarding the two sensitivities, and we thank the Reviewer for pointing this out.

We revised the abstract and clarified the two sensitivity concepts. For the dual-comb results, we wrote:

“... we **measure** high-resolution broadband (>2 THz) overtone spectra for acetylene gas **and obtain a normalized noise equivalent absorption coefficient of $1.71 \times 10^{-11} \text{ cm}^{-1} \cdot \text{W} \cdot \text{Hz}^{-1/2}$** .” (p. 2, line 10; original line 29) Here, we also merged the advice from Reviewer 2 for using the normalized noise equivalent absorption (NNEA, $1.71 \times 10^{-11} \text{ cm}^{-1} \cdot \text{W} \cdot \text{Hz}^{-1/2}$) instead of 1.1 ppb.

For describing the detection limit of the optomechanical resonator, we wrote:

“The optomechanical resonator, with a detection limit down to 15 parts per trillion, allows broadband dual-comb excitation.” (p. 2, line 12; original line 30)

Indeed, the sensitivities in the dual-comb and the Allan-Werle measurement (Fig. 4 e) were consistent. In the Allan-Werle measurement, we turned on the CW excitation laser of 200 mW (the maximum power) and obtained a sensitivity limit down to 24 ppt at 100 s (see Fig. 4 e, **Laser on**), which was slightly above the sensor’s detection limit of 15 ppt, indicated by the Laser-off trace (Fig. 4e), because the CW laser induced thermal noise. In the dual-comb measurement, the 1.1 ppb at 100 s was achieved with a total power of 150 mW or 3.75 mW (=150 mW/40) per dual-comb line. Suppose one increases the power per dual-comb line to 200 mW (equal to the CW laser). In that case, the sensitivity will be improved by 53 (=200mW/3.75mW) times as about 21 ppt at 100 s, provided the photoacoustic signal linearly depends on the excitation power (see Fig. 4d).

BANDWIDTH

The simultaneous bandwidth shown in this work (2 nm/240 GHz) can be achieved without any problems with conventional cavity-enhanced comb-based techniques with the comb locked to the cavity. Even 40 nm (the entire tuning range of the comb used in this work) can be simultaneously transmitted by commonly available near-infrared mirrors. The total bandwidth of 2.1 THz demonstrated here is obtained by stitching 13 spectra, which is hardly optimum or desired for real-time broadband sensing. It increases the measurement time by a factor of 13, not counting the dead time needed to tune the laser.

A discussion about the possible extension of the simultaneous bandwidth of this approach must be added, and the limitations should be discussed. The authors should at least calculate how large optical bandwidth one can fit into the bandwidth of the MIM system depending on the choice of f_{rep} and Δf_{rep} .

In this approach, the cavity needs to be resonant for the probe laser, but not for the comb. A discussion about the limitation to this approach is needed, considering mirror coatings or the choice of probe and comb laser wavelengths.

We agree with the Reviewer that the simultaneous bandwidth as well as the entire spectral range (with spectral stitching) in our work can be achieved with other conventional cavity-enhanced techniques. But we should point out two things.

First, the simultaneous bandwidth is currently limited by the laser source we used, i.e., the dual electro-optic (EO) combs, instead of the concept we proposed or the optomechanical sensor. The laser source is not a fundamental limitation of our method and can be overcome with broadband combs. Highly-coherent mode-locked combs, with simultaneous bandwidth over 200 nm in both near-infrared (1.5 μm) and mid-infrared (3-14 μm) regions, are commercially available (see <https://www.menlosystems.com/>). Broadband comb, simultaneously spanning, e.g., from 350 nm to 22.4 μm [*Nat. Photon.* **15**, 281 (2021)] (cited as ref. [38]), has also been achieved by Daniel M. B. Lesko et al. Although we currently have no access to these combs, such combs will undoubtedly improve the spectral width of our system.

Here, we used EO combs not only because they were accessible to us but also because they offered excellent mutual coherence without additional phase-locking loops and could be tuned for extending the spectral range (as we stated in the experimental setup section). This type of EO comb has been commonly used for verifying novel dual-comb concepts. For instance, similar EO combs have been employed in the first demonstration of photoacoustic dual-comb spectroscopy [*Nat. Commun.* **11**, 3152 (2020)] (ref. [19]), dual-comb photothermal spectroscopy [*Nat. Commun.* **13**, 2181 (2022)] (ref. [21]), dual-comb hyperspectral imaging [*Optica* **7**, 199 (2020)], dual-comb digital holography [*Nat. Photon.* **15**, 890 (2021)], and so on.

Second, the bandwidth of conventional cavity-enhanced comb spectroscopy depends on the mirror coating technology. Currently, the technical problem of broadband mirror coating is difficult to tackle, causing a trade-off between the spectral bandwidth and the cavity finesse for sensitivity enhancement. Our method bypasses this technical problem since the cavity is only resonant with the single-mode probe laser. Thus, our concept opens new opportunities for broadband measurement.

We admit that our goal is “*real-time broadband sensing*”, but the system is currently limited by the EO combs. Therefore, in this article we only claimed “real-time multiplexed sensing” (p. 4, line 17; original line 72) and “ultrasensitive multiplexed spectroscopy” (p. 5, line 4; original line 82), both of which have been supported with experimental data (Fig. 3 and Fig. 4). After all, we demonstrated the unprecedented multiplexing capability for cavity-enhanced **photoacoustic spectroscopy**.

We are grateful to the Reviewer for advising us to add a discussion. *A discussion about the possible extension of the simultaneous bandwidth of this approach must be added, and the limitations should be discussed. The authors should at least calculate how large optical bandwidth one can fit into the bandwidth of the MIM system depending on the choice of f_{rep} and Δf_{rep} .*

This suggestion indeed improves our manuscript. Following the suggestion, we added a relevant discussion in the revised manuscript.

“In contrast to conventional cavity-enhanced spectroscopy, our method bypasses the limitation on spectral bandwidth imposed by mirror coatings (Supplementary Note 4). The maximum spectral bandwidth ($\Delta\nu = \Delta B \cdot f_r / \Delta f_r$), determined by our acoustic bandwidth ($\Delta B = 340$ kHz within 10 dB), is 68 THz for $f_r = 400$ MHz and $\Delta f_r = 20$ Hz and could be enlarged by optimizing $f_r / \Delta f_r$. Currently, the EO combs limit the spectral bandwidth of our system, which, however, can be overcome by using highly-coherent broadband combs^{36,37}.” (p. 17, line 13; original line 301)

We added two references regarding broadband combs.

[36] Lesko, D.M.B. et al. A six-octave optical frequency comb from a scalable few-cycle erbium fibre laser. *Nat. Photonics* **15**, 281–286 (2021).

[37] Carlson, D. R. et al. Ultrafast electro-optic light with subcycle control. *Science* **361**, 1358–1363(2018).

The acoustic bandwidth in our system is 340 kHz within 10 dB (1.1 MHz for a full span), which

eventually limits the simultaneous spectral bandwidth of our approach. The maximum simultaneous spectral bandwidth, $\Delta\nu$, can be expressed as: $\Delta\nu = \Delta B \cdot f_r / \Delta f_r$, where ΔB is the acoustic bandwidth. For example, for $f_r = 400$ MHz and $\Delta f_r = 20$ Hz (the parameters used in the article), we obtain $\Delta\nu = 68$ THz (2270 cm^{-1}) or ~ 500 nm around 1550 nm, provided $\Delta\nu$ is within the comb spectral width.

We also agree with the Reviewer that *“In this approach, the cavity needs to be resonant for the probe laser, but not for the comb. A discussion about the limitation to this approach is needed, considering mirror coatings or the choice of probe and comb laser wavelengths.”*

We measured the spectral transmittance of the mirror-in-the-middle (MIM) cavity, as shown in Fig. R1. The spectrum spans from 1200 nm to 4500 nm. The cavity is designed to be resonant with the probe laser at 1064 nm, which explains the cutoff at 1200 nm. Currently, the cavity mirrors are not for mid-infrared wavelengths above 4500 nm.

Fig. R1 Transmission spectra of the resonant cavity. The spectra are measured with a commercial infrared spectrometer (spotlight400, PerkinElmer; resolution of 1 cm^{-1}).

A possible solution is to separate the excitation and probe beams spatially. The excitation beam spatially coincides with the probe on the MIM membrane but does not pass through the resonant mirror, as depicted in Fig. R2. In our experiment, we noticed that making the excitation and probe beams propagate collinearly was unnecessary. As long as the molecules around the MIM membrane were excited, we could observe the interferometric signal sensed by the probe beam.

Fig. R2 Configurations of DCOS. a, collinear configuration; b, DCOS with crossed beams.

We added the figures (Figs. R1 and R2) as **Supplementary Figs. 4 and 5** and a description in the revised supplementary materials (see **Supplementary Note 4**).

SENSITIVITY

The 1.1 ppb detection sensitivity at 100 s calculated from the SNR of the molecular signal is the limit that should be stated in the abstract. The authors instead state a 15 ppt detection limit, which I believe is calculated incorrectly and is not supported by the data. See comments under ULTRASENSITIVE DETECTION section below. For the 15 ppt detection limit, the authors characterized the noise in the system under different conditions than used for molecular detection, and it is not clear how they would justify the 100-fold improvement.

We thank the Reviewer for allowing us to explain the 15-ppt detection limit. This is the detection limit of our optomechanical sensor. We performed Allan-Werle deviation measurement following the standard procedure for characterizing photoacoustic sensors [*Rev. Sci. Instrum.* **90**, 023102 (2019); *Nat. Commun.* **8**, 15331 (2017)]. The procedure has been given in Methods. The 15-ppt detection limit was supported by the data (Laser off) in Fig. 4e. Compared to this detection limit (15 ppt), the dual-comb result (1.1 ppb) was nearly 100-fold worse due to the limited power per comb line.

Following the Reviewer's advice "*The 1.1 ppb detection sensitivity at 100 s calculated from the SNR of the molecular signal is the limit that should be stated in the abstract.*" We revised the abstract. It now reads, "... we measure high-resolution broadband (>2 THz) overtone spectra for acetylene gas and obtain a normalized noise equivalent absorption coefficient of $1.71 \times 10^{-11} \text{ cm}^{-1} \cdot \text{W} \cdot \text{Hz}^{-1/2}$." We used the unit ($\text{cm}^{-1} \cdot \text{W} \cdot \text{Hz}^{-1/2}$) which is normalized to laser power and measurement time and is more suitable for general comparison (also suggested by Reviewer 2).

To achieve the 1.1 ppb detection limit, all comb power was channeled into very few spectral components (150 mW and >40 comb lines, spaced by 700 MHz). For a broader comb, the sensitivity will get worse, roughly by the amount of broadening, if the total power stays the same. Thus, this technique does not combine a wider bandwidth with the 1.1 ppb sensitivity.

The Reviewer pointed out a general tradeoff between the bandwidth and the sensitivity for any spectrometer or spectral sensor.

Thanks to the combination of DCS and optomechanics, our system is advantageous respecting this tradeoff. For instance, broadband gas sensing with high sensitivity is demonstrated by *Nat. Commun.* **13**, 2181 (2022) (ref. [21]) where a spectrum of 1 THz (stitching 15 spectra) and a sensitivity of 8.7 ppm within 40-GHz optical bandwidth and 1000-s integration time are obtained. Our results showed a broader spectral range (2.1 THz, with stitching 13 spectra) and nearly four orders of magnitude sensitivity improvement (1.1 ppb with 30-GHz bandwidth in 100 s). Also, state-of-the-art cavity-enhanced photoacoustic systems have NNEA varying from 10^{-8} - 10^{-12} $\text{cm}^{-1}\cdot\text{W}\cdot\text{Hz}^{-1/2}$ for single spectral element only. We obtained $\text{NNEA}=1.71\times 10^{-11}$ $\text{cm}^{-1}\cdot\text{W}\cdot\text{Hz}^{-1/2}$ for 40 elements measured simultaneously. More comparisons can be found in the tables in the manuscript and supplementary materials.

The C₂H₂ lines detected in this work come from a combination band that is particularly strong in the near-infrared, with line strengths of the order of 10^{-20} cm/molec. The ‘typical’ line strengths in the MIR are 10^{-19} to 10^{-20} cm/molec, e.g. for methane at 3.3 μm , so the absorption sensitivity improvement will be at most one order of magnitude, not 2-3 as stated in the conclusions.

We agree with the Reviewer that “The ‘typical’ line strengths in the MIR are 10^{-19} to 10^{-20} cm/molec”, although there are a few important environmental gases having strong absorptions in the MIR. For example, the line strength for CO₂ reaches 3.54×10^{-18} cm/molecule (at 2361.46 cm^{-1} or 4.23 μm). Carbonyl sulfide (COS) has strong absorptions (line strength of 1.25×10^{-18} cm/molecule) at 2070.8 cm^{-1} or 4.83 μm .

Following the Reviewer’s comments, we modified the sentence to:

“Furthermore, accessing the mid-infrared region, where the absorption line strengths of fundamental transitions are typically **one to two orders** (p. 17, line 10; original line 299) of magnitude stronger than that of the overtones in the near-infrared, will improve the sensitivity, possibly down to the **ppt** regime.” (p. 17, line 12; original line 301)

COMPACTNESS

Compactness is a subjective criterion, but if the authors want to claim that their system is particularly compact, they should discuss the physical size of the device or show a photograph. Considering the simplicity of the approach, the authors point out that the cavity does not need to be resonant for the comb; however, a cavity with PDH locking is needed for the MIM system.

We thank the Reviewer for this critical advice. We have no intention of emphasizing the compactness of the current setup in this proof-of-concept demonstration. After all, the combs and the PDH system are not minimized. But, as a prospect, we pointed out that our concept could lead to a miniaturized dual-comb spectrometer with high sensitivity, considering the advancement of micro-sized combs (ref. [38]) and optomechanical sensors (ref. [32]). Again, this is a prospect (the last paragraph in Discussion).

The Reviewer suggested us to add a picture of our MIM system (the gas sensor) in the revised supplementary materials Supplementary Fig. 7. The picture is also shown in the following. The size is $3 \times 3 \times 4.9 \text{ cm}^3$. Since our system does not need a long interaction length between the light and molecules, the size of our sample cell can be relatively small.

Fig. R3 Picture of the MIM system.

Our method is simple relative to the case of dual-comb cavity-enhanced spectroscopy, which locks a cavity to optical combs. Our concept supports comb-based molecular spectroscopy with cavity-enhanced sensitivity, where the cavity is stabilized to a CW laser. Considering the maturity of PDH locking technique, we did not recognize it as a difficulty.

OTHER ISSUES

The authors should discuss the limitation on the sample pressure; does the sensor operate only at atmospheric pressure? If yes, this will limit the applicability of this approach for multispecies detection because of spectral overlap of lines of different species.

We thank the Reviewer for the comments, which are valuable to improve our manuscript. Our system can operate at lower pressure for Doppler-limited high-resolution spectral measurements.

Following the suggestion, we added a supplementary note (Note 3. High-resolution measurement) with the following figure (as Supplementary Fig. 3). We also added a discussion of the limitation on the sample pressure in the main text. It reads,

“We also measure Doppler-limited high-resolution spectra under gas pressure of 10^3 Pa with $f_r^{(1)}=100 \text{ MHz}$ (see Supplementary Note 3). Note that, like any photoacoustic sensor¹²⁻²⁰, our system detects pressure waves, the strength of which decreases with reduced gas pressure.” (p. 13, lines 16-19; original line 229)

Fig. R4 Spectral measurements for the $(\nu_1+\nu_3)$ band P(17) line of $^{12}\text{C}_2\text{H}_2$ (a) at pressure of 10^3 Pa and (b) under atmospheric pressure.

We measured molecular spectra at 10^3 Pa with our setup for the $(\nu_1+\nu_3)$ band P(17) line of 50% $^{12}\text{C}_2\text{H}_2$. The results in Fig. R4 (a) measured within 2 s are shown in the following. The dual-comb line spacing is set to 100 MHz for measuring the Doppler-limited linewidth. The data shows a linewidth of 520 MHz, close to the Doppler limit (470 MHz). The collision broadening caused by gas molecules hitting the membrane may explain the difference. The results measured under atmospheric pressure (Fig. R4(b)) are displayed for comparison.

It is not clear how the signal is extracted from the PDH error signal. In closed loop, i.e. under locked conditions, at frequencies within the locking bandwidth, the error signal should be zero. Is the bandwidth of the lock much lower than 70 kHz, the frequency at which the interferograms are detected? If not, how can the signal be observed there? See more comments under NOTE 4 below.

In our experiment, the PDH error signal is feedback to the probe laser through the PZT of the laser rather than the current of the laser. Therefore, the feedback bandwidth should be much lower than 70kHz. To support this claim, we measured the closed loop's frequency response, as shown in Fig. R5. The frequency response of the closed loop is normalized with an open loop signal (with an additional probe light field which is not in resonant with the optical cavity and thus without the influence of feedback). Figure R5 shows the frequency response of PDH locking from 1 kHz to 200 kHz. The 3-dB frequency is around 45 kHz, which is much lower than 70 kHz. Therefore, the signal at ~ 70 kHz is barely influenced by the PDH feedback loop.

Fig. R5 Frequency response of the closed loop.

English must be fixed at quite a few places.

We have carefully revised our manuscript and modified the English. The changes are given in detail by replying to the following comments.

BOTTOM LINE

The conclusions in this paper are not supported by the data. In particular, the authors claim

- *Broad spectral coverage of 2 THz – but this is achieved by stitching of 13 spectra; the simultaneous bandwidth is 2 nm (240 GHz), and even smaller (40 x 700 MHz = 28 GHz) for the data used to evaluate the detection limit (i.e. to achieve the highest SNR).*
- *Millisecond measurement time – it is indeed demonstrated, but the 1.1 ppb sensitivity is achieved for 100 s acquisition time, and it will be 300 times worse at 1 ms.*
- *Detection limit of 15 ppt – this is wrong, the limit is 1.1 ppb at 100 s, see above and below.*

We thank the Reviewer for allowing us to clarify the issues of our conclusions.

First, the broad spectral coverage of 2 THz means our current system can measure a spectrum spanning over 2 THz. As far as we understand, “spectral coverage” and “simultaneous bandwidth” are two different concepts. Despite being willing to achieve simultaneous bandwidth of >2THz, we are limited by the EO comb source we used. This is a common issue for demonstrations with EO combs.

Second, following the comments “*the 1.1 ppb sensitivity is achieved for 100 s acquisition time, and it will be 300 times worse at 1 ms*”, our system would have a sensitivity of 330 ppb (=1.1 ppb×300) for a 1-ms measurement. This is a significant improvement compared to 8.7 ppm at 1000s [*Nat. Commun.* **13**, 2181 (2022)] and 10 ppm at 1000 s [*Nat. Commun.* **11**, 3152 (2020)]. Multiplexed photoacoustic sensing with short measurement times and high sensitivity remains challenging. Our method takes a step towards this direction.

Third, we are sorry the way we described the sensitivity was misleading. The sensitivity measured with the combs was 1.1 ppb, and the detection limit of the sensor was 15 ppt.

To avoid misunderstanding, we modified our conclusion as follows:

“Our method, unifying the two revolutionary techniques —DCS and cavity optomechanics, is promising for spectral measurement with broad spectral coverage, high resolution, short measurement times, and most importantly ultrahigh sensitivity.” (p. 18, Conclusion)

INTRODUCTION

1. Line 48: Add references in the last sentence of this paragraph.

We added references [1], [2] and [3] to the sentence as:

“...selective multispecies detection³, reliable analysis of complex mixtures¹, and real-time monitoring of trace gases².” (p. 3, lines 13-14; original line 48)

2. Line 49: The first sentence needs rephrasing, ‘Ultrasensitive DCS has been investigated..’

We rephrased the sentence as:

“Ultrasensitive DCS has been demonstrated using hollow-core fibers⁷, multi-pass cells⁸, and optical resonance cavities⁹⁻¹¹.” (p. 3, line 15; original line 49)

3. Line 53: Provide references to works that achieve ppt sensitivities.

We added ref. [10] (Sulzer, P. et al. Cavity-enhanced field-resolved spectroscopy. *Nat. Photon.* **16**, 692-697 (2022)) to the sentence, “Among these demonstrations, cavity-enhanced schemes, with unmatched pathlength enhancement, yield the lowest detection limits — possibly down to the parts-per-trillion (ppt) level¹⁰.” (p. 3, line 18; original line 53)

4. Line 54: The phrasing ‘respecting mirror coating, comb-cavity coupling, dispersion’ is not clear.

We modified the sentence as:

“The resonance cavities, however, essentially restrict the wavelength range of operation due to the difficulty of broadband mirror coating. Also, cavity-enhanced DCS needs extra efforts for comb-cavity coupling⁹ and intracavity dispersion control^{10, 11}.” (p. 3, lines 18-21; original lines 53-55)

5. Line 68: ‘limit overall performances’ – please specify what performance parameter you are referring to.

We modified the sentence to:

“Also, the narrow acoustic bandwidths of the cavity-enhanced photoacoustic (PA) systems (e.g. 1 Hz in Ref. 15) limit their overall performances (such as acquisition speeds and spectral widths).” (p.

4, line 12; original line 68)

6. Line 73: *demanding* → *demanded*.

We changed “demanding” to “demanded”. (p. 4, line 18; original line 73)

BASIC PRINCIPLE

7. Fig. 1a: It is not clear from this figure how the comb light is coupled to the cavity (from which direction). Please draw it.

We draw the direction of the coupled combs in the revised Fig. 1a.

8. Line 117: Specify what ‘considerably small’ means. Small with respect to what?

We modified the sentence to:

“Suppose $f_{\text{OF}(n)}$ matches a molecular ro-vibrational transition and Δf_0 and Δf_r are considerably small with respect to $f_{\text{OF}(n)}$ ” (p. 7, line 10; original line 117)

EXPERIMENTAL SETUP

9. Line 163: For clarity, add: mirrors ‘resonant for the probe laser’.

We added “For ultrasensitive PA detection, a MIM system, consisting of two plano-concave mirrors resonant for the probe laser” (p. 9, line 25; original line 163)

10. Line 172: It is stated here that the comb beam is aimed at the geometric center of the membrane, but it should also be stated how it is coupled to the cavity (through the mirror? at an angle?).

In the revised manuscript, we wrote, “For spectroscopic sensing, we shine the dual-comb beam on the geometric center of the membrane. As shown in Fig. 2a, the dual-comb beam is coupled into the cavity through a cavity mirror at near normal incidence.” (p. 10, line 8; original line 172)

11. Line 180: Reconsider the use of ‘inhibiting’, do you mean cancellation?

Exactly. In the revised version, we changed “inhibiting” to “cancelling.” (p. 10, line 17; original line 180)

12. Line 184: *Originated* → *originating*.

We changed “originated” to “originating.” (p. 10, line 21; original line 184)

13. Line 187: *Locate* → *fall*.

We changed “locate” to “fall.” (p. 11, line 1; original line 187)

MULTIPLEXED SPECTRAL MEASUREMENT

14. Line 211: The PA spectrum should be baseline free, but here the authors discuss removing the baseline. This is contradictory and the origin of this baseline needs to be clarified.

The PA spectrum is baseline free. However, the spectral profile of the excitation combs and the spectral response of the MIM sensor are not flat, leading to a spectral background that should be considered. Here, “*removing the baseline*” means removing the spectral background of the combs and the sensor.

We notice that the phrase, “baseline”, may be misleading. We modified the sentence to:

“We also remove the spectral background of the combs and the MIM sensor by using the PA spectrum divided by the convolution of the membrane response curve and the dual-comb spectral outline.” (p. 12, line 2; original line 211)

15. Fig. 3b: Discuss the origin of the spikes next to the stronger line around 68-69 kHz.

We added a sentence in the caption of Fig. 3 to explain the spikes. The sentence reads:

“Also, the spikes around 68-69 kHz are caused by the high-intensity lines at the center of the EO combs (as shown in supplementary Fig. 1).” (p. 13, line 4; original line 218)

16. Line 221, 228, and 316: 400 MHz is the sample point spacing, not the resolution.

In the revised manuscript, we changed them to sample point spacing. (p. 13, line 7, 15; original line 221,228)

ULTRASENSITIVE DETECTION

17. Line 240 and 248: It is not clear if the SNR is calculated on the peak of the absorption line or of a comb line. Clarify. It should be taken on the absorption line.

The SNR is obtained for the peak of the absorption line.

We added the above sentence in the revised manuscript (p. 14, line 17; original line 249).

18. Fig. 4b: There are visible systematic discrepancies between the HITRAN model and the data. Plot the residuals and discuss the cause of the discrepancies.

We plotted the residuals in the revised version, as shown in the following. We also updated Fig. 4b in the revised manuscript.

Fig. R6 The updated Fig. 4b in the revised manuscript.

The discrepancies are within $\pm 8\%$ between the 2-s data and the HITRAN simulation. The discrepancies are caused by the additional electronic amplifier we used for amplifying the small PA signal in the low-concentration measurement. The amplifier may distort weak components at the edges. Since the discrepancies are relatively small, they do not significantly influence concentration measurements. For high-concentration measurements (see Fig. R4(b)), we did not notice the discrepancies around 195.258 THz.

We added the following sentence in the caption part of Fig. 4b.

“The systematic discrepancies around 195.258 THz are due to the additional electronic amplifier we used for amplifying the PA signal in the low-concentration measurement.”

19. Line 253: It is stated that there is an ‘orders of magnitude sensitivity improvement’ with respect to previous comb-based PA works. However, from table 1 in the supplementary it is clear that the improvement is less than 2 orders of magnitude. Rephrase this conclusion in the main text.

In the revised paper, the rephrased sentence reads:

“The results manifest nearly two orders of magnitude sensitivity improvement compared to the existing comb-enabled PAS¹⁶⁻²⁰ and photothermal systems²¹ (see Table 1).” (p. 14, line 20; original line 253)

20. Line 257: Add reference to Werle’s paper introducing the Allan-Werle method. It is Allan, not Allen.

Following the Reviewer’s suggestion, we add a new reference as ref. [34] [35] and corrected “Allen” to “Allan”. (p. 15, line 1; original line 257)

[34] Wu, H. et al. Beat frequency quartz-enhanced photoacoustic spectroscopy for fast and calibration-free continuous trace-gas monitoring. *Nat Commun* **8**, 15331 (2017).

[35] Zhao, P. et al. Mode-phase-difference photothermal spectroscopy for gas detection with an anti-resonant hollow-core optical fiber. *Nat Commun* **11**, 847 (2020).

21. Line 258: What does 'routinely' mean here?

Since we added ref. [34] introducing the Allan-Werle method in the revised paper, we removed the word "routinely". (p. 15, line 2; original line 258)

22. Line 257 and on: It is not clear what CW laser is used here. The methods section, line 359, states that it is 'the same' laser. Same as what? Is that the seed laser of the EO combs? Specify.

It is the same laser used for the seed of the EO combs. To make it clear, we modified the sentence to:

"... we use the intensity-modulated cw laser (the same one for seeding the EO combs) as the excitation beam..." (p. 21, line 18; original line 360)

What is the amplitude of the applied intensity modulation compared to the amplitude of the interferogram signal? Allan-Werle deviation must be measured under the same conditions as the molecular signal from the sample. Here, it seems the authors characterized the noise in the system under different conditions. For example, they used a lock-in amplifier for the Allan-Werle plot, but not with the sample. How is the calibration coefficient (line 364) obtained? How is the voltage output of the lock-in amplifier compared to the interferogram amplitude?

The signal generated by a CW laser with intensity modulation is much stronger than the dual-comb interferogram signal because the powers are different (despite the total power being the same). The power for the CW case is 150 mW, and the power per comb line, which matters to the photoacoustic signal, is $150/40=3.75$ mW.

We performed the Allan-Werle analysis due to two considerations: 1) to confirm the thermal effect and the detection limit of the MIM sensor (as we have clearly stated in the manuscript); 2) this is a standard process for characterizing a photoacoustic sensor and can be used for direct comparison with other works. In our work, we obtained the dual-comb sensitivity of 1.1 ppb at 100s, which is directly comparable to other dual-comb systems, and we also obtained the detection limit from the Allan-Werle analysis which is directly comparable to other CW-based photoacoustic sensors or methods.

The Reviewer pointed out that "*For example, they used a lock-in amplifier for the Allan-Werle plot, but not with the sample.*" In fact, this is a standard measurement (measured under pure N₂) in photoacoustic spectroscopy (see *Nat Commun* **8**, 15331 (2017), also ref. [33]). We thought that it would be helpful to add this measurement because it offered different information about our MIM system, as we have explained before.

The calibration coefficient is obtained using a targeted gas (i.e., C₂H₂), the details of which we have

described in Methods. We also added two relevant references as ref. [34] and [35].

How can the authors explain the 15 ppt limit at 110 s, compared to 1.1 ppb at 100 s using the SNR method? What is the reason of this discrepancy of two orders of magnitude? I strongly suspect the 15 ppt detection limit is wrong.

As we have explained before, if one rules out the laser power factor, the two measurements yield similar results. Our measurements are consistent.

DISCUSSION

23. Line 292 and on: The authors derive that the detection limit is 10 times above the shot noise. It is not entirely clear which of the detection limits they refer to (the 1.1 ppb or 15 ppt).

Here, we measured the displacement sensitivity of our MIM system, which was $1.2 \text{ fm}/\sqrt{\text{Hz}}$. This number was “10 times above the shot noise.”

In the original manuscript, we wrote, “Furthermore, we compare our MIM system to a Michelson interferometer with the same displacement signal and find that the displacement sensitivity of our detection system is $1.2 \text{ fm}/\sqrt{\text{Hz}}$, which is within 10 dB of the shot noise limit (see Supplementary Note 5).”

SUPPLEMENTARY

TABLE 1: The spectral bandwidth of 70 cm^{-1} cannot be listed together with the NNEA of $1.7e-11 \text{ cm}^{-1} \text{ W Hz}^{-1/2}$, because the 70 cm^{-1} bandwidth requires stitching of 13 spectra. The authors should either write the bandwidth of a single spectrum in the table, or recalculate the NNEA by increasing the measurement time by 13.

We revised Table 1 using 70 cm^{-1} (stitching 13 spectra) instead of 70 cm^{-1} . The spectral widths of previous works were also replaced with the simultaneous bandwidths.

TABLE 2: In this table, the authors should list the demonstrated detection limit of 1.1 ppb, not 0.015 ppb (they actually write 0.027 ppb – not sure where this number comes from). According to this table, the current work does not supersede the sensitivity of CW based CE demonstrations.

In order to directly compare the sensitivity achieved by optical combs with that of a CW laser, we considered the total power (P) being divided by the number of comb lines (N). In our case, $P=150 \text{ mW}$, $N=40$, and the sensitivity is 1.1 ppb. That means one can achieve 0.027 ppb for $P=150 \text{ mW}$ and $N=1$, provided the photoacoustic signal linearly depends on the excitation power.

We agree with the Reviewer that when describing the sensitivity, we should use 1.1 ppb because we already wrote “40 elements” in the “Spectral elements in a single measurement” column in Supplementary Table 1 (Supplementary Table 2 in the original version). Therefore, we replaced 0.027 ppb to 1.1 ppb in the revised version.

We agree that ‘*the current work does not supersede the sensitivity of CW based CE demonstrations.*’ However, simply comparing the sensitivity could be biased.

First, our current system provided 40 spectral elements in a single shot. This feature is not what a CW-based CE system could provide.

Second, our sensitivity could be improved by increasing the comb power with additional fiber amplifiers. After all, the limiting factor of our current sensitivity is the power per comb line.

Third, a single gas sensing method with the capabilities of multiplexed or broadband spectral measurement, fast measurement, and ultrasensitive measurement is already of importance to many applications, e.g., breath diagnostics (ref. [1]) and environmental monitoring (ref. [2] and [3]), even though these features may not be their optimum simultaneously.

NOTE 4:

It seems that the entire derivation is for an open loop error signal. It is not clear how the molecular signal is extracted in closed loop, under locked conditions. Its amplitude will depend on the gain in the feedback loop, this must be discussed.

The frequency of the spectral signal we measured is larger than the feedback bandwidth, therefore, the derivation can be treated for an open loop error signal. We have added the discussion in the revised Supplementary Note 6.

- *For Eq. (1), cite the original paper by Bjorklund instead of ref. S16.*

We have cited the paper by Bjorklund instead of ref. S11 in the revised Supplementary Materials.

- *Above Eq. (4): ‘overlap coefficient’ – overlap of what?*

The overlap coefficient depends on the spatial overlap between of the mechanical and the optical spatial modes. In our experiment, we used the Gaussian cavity mode. In the Note 4 (Note 6 in the revised version), we assume that the mechanical spatial mode matches the optical spatial mode perfectly, and overlap coefficient is approximately 1. We have added this part in Supplementary Note 6.

- *Above Eq. (5): State the LO frequency.*

The LO frequency is the local oscillator frequency, which is 3MHz as used in the experiment. We stated this LO frequency in the revised version.

- *Fig. 4 and below: Specify the pressure and power at which this dependence was derived. I see that these parameters are written in Eq. 11, but for clarity they should be stated together with the result of the calculation.*

NCOMMS-23-10200

The pressure and power have been specified in Supplementary Note 6.

Reviewer #2 (Remarks to the Author):

The authors have addressed my comments from the previous round of review and I would recommend the manuscript for publication.

Reviewer #3 (Remarks to the Author):

The authors have improved the manuscript following my and the other reviewer's comments. However, the discussion about the sensitivity is still misleading in the paper. In the response letter, the authors clearly distinguish between the detection limit of DCOS and the sensor itself, but in the manuscript this distinction is not clear. This issue must be addressed before the paper is published. I also have a few smaller comments appended at the end of this letter.

In the response letter, the authors write:

'There are two concepts of sensitivity in our manuscript; one is the sensitivity measured with the combs (i.e., 1.1 ppb at 100 s for 40 spectral elements), and the other is the detection limit of the optomechanical sensor (i.e., 15 ppt at 110 s based on the intrinsic noise of the sensor). The latter was obtained from a standard Allan-Werle deviation measurement (Fig. 4 e, Laser off) when the excitation laser was switched off.'

and

'We performed the Allan-Werle analysis due to two considerations: 1) to confirm the thermal effect and the detection limit of the MIM sensor (as we have clearly stated in the manuscript); 2) this is a standard process for characterizing a photoacoustic sensor and can be used for direct comparison with other works. In our work, we obtained the dual-comb sensitivity of 1.1 ppb at 100s, which is directly comparable to other dual-comb systems, and we also obtained the detection limit from the Allan-Werle analysis which is directly comparable to other CW-based photoacoustic sensors or methods.'

These descriptions are clear, but they are nowhere to be found in the manuscript. In the manuscript, it is clear that the first detection limit is determined from the SNR of the P(17) line measured at 1 ppm concentration using 150 mW of total comb power distributed over 40 spectral elements (comb modes). Then, the authors measure the noise in the same system filled with N₂ using a CW laser with similar total power (200 mW) and without the excitation laser. It is shown that the CW laser does not induce thermal noise (Fig. 4e). However, what is not clear is how the signal measured with the CW laser was converted to concentration. In the response letter, the authors write: 'The calibration coefficient is obtained using a targeted gas (i.e., C₂H₂), the details of which we have described in Methods.' In the Methods section, line 389, it is stated that 'For the plots in Fig. 4e, the voltage signal is converted to the equivalent gas concentration, using a measured coefficient of 305.2 uV/ppm for 12C₂H₂.' It is not explained anywhere HOW this coefficient was measured, while it is the key to the conversion of the measured noise to concentration detection limit. The authors must explain that. I assume that this coefficient is obtained from a calibration measurement of C₂H₂ using the CW laser with 200 mW of power. This coefficient would be different if it was determined using the comb instead (since the power is distributed over the comb lines).

The authors should clearly distinguish between the two detection limits in the paper, they should state that the second, lower, detection limit is for a single element detection only, and that it is presented to enable comparison with other CW-based PAS methods.

Moreover, Reviewer 2 pointed out in their first comment that expressing the detection limit in ppb/ppm units is highly dependent on the selected gas etc. This comment applies also to the detection limit of the sensor itself. The authors write in their response letter:

'We highlighted the 15-ppt in the abstract because it represented the intrinsic characteristics of

our optomechanical sensor and was essential for comparison with other photoacoustic sensors. In contrast, the 1.1-ppb was the sensitivity of our current dual-comb setup, which was affected by, e.g., the excitation comb power and the molecular line strengths.'

This does not make sense, because the 15 ppt limit is also affected by the excitation power and the molecular line strength; it is specific to the particular molecular line chosen. Therefore, the sensor detection limit should also be expressed in units that are independent of the gas, either in the units of absorption coefficient (cm⁻¹), units of NNEA, or in fm/sqrt(Hz).

In the letter, the authors discuss that the two detection limits are consistent:

'The sensitivities in the dual-comb and the Allan-Werle measurement (Fig. 4 e) were consistent. In the Allan-Werle measurement, we turned on the CW excitation laser of 200 mW (the maximum power) and obtained a sensitivity limit down to 24 ppt at 100 s (see Fig. 4 e, Laser on), which was slightly above the sensor's detection limit of 15 ppt, indicated by the Laser-off trace (Fig. 4e), because the CW laser induced thermal noise. In the dual-comb measurement, the 1.1 ppb at 100 s was achieved with a total power of 150 mW or 3.75 mW (=150 mW/40) per dual-comb line. Suppose one increases the power per dual-comb line to 200 mW (equal to the CW laser). In that case, the sensitivity will be improved by 53 (=200mW/3.75mW) times as about 21 ppt at 100 s, provided the photoacoustic signal linearly depends on the excitation power (see Fig. 4d).'

This discussion is correct, but to achieve 200 mW per comb line, while having the same number of comb modes (40), would require 8 W of power in the 30 GHz bandwidth. The authors should include the above discussion in the paper, and comment on what kind of laser they suggest using at these power levels. Will the sensor operate at such high incident power, won't thermal noise be induced then, etc.?

Concerning the Allan-Werle analysis: the authors should cite the original paper where the method was introduced, P. Werle et al, Appl. Phys. B 57, 131–139 (1993). The references that the authors cite now, Ref. 34, Wu et al, and 35, Zhao et al, are also relevant because they show the use of the method in other systems. I note that in both these references the Allan-Werle plot is measured under the same illumination conditions as the sample signal was recorded, but with the sensor filled with N₂. In this work, the authors have changed the illumination conditions from dual-comb to continuous wave.

The phrasings that must be changed in the paper are:

1. Line 29: 'we measure high-resolution broadband (>2 THz) overtone spectra for acetylene gas and obtain a normalized noise equivalent absorption coefficient of $1.71 \times 10^{-11} \text{ cm}^{-1} \cdot \text{W} \cdot \text{Hz}^{-1/2}$.' It sounds as if these were obtained simultaneously, while they are not. The NNEA is evaluated in 30 GHz bandwidth. The NNEA would be different when calculated over the 2 THz bandwidth. I suggest saying '... and obtain a normalized noise equivalent absorption coefficient of $1.71 \times 10^{-11} \text{ cm}^{-1} \cdot \text{W} \cdot \text{Hz}^{-1/2}$ in a spectrum with 30 GHz simultaneous bandwidth.'
2. Line 31: 'The optomechanical resonator, with a detection limit down to 15 parts per trillion, allows broadband dual-comb excitation.' This is misleading as it suggests that the 15 ppt limit is obtained under broadband dual comb excitation. It must be stated that this limit is obtained under CW operation, and it should be expressed in other units.
3. Line 270: When starting the discussion about the second detection limit, the authors write: 'To confirm the detection limit of our system', which suggests that they continue discussing the same detection limit, which is not the case. Instead, this paragraph should start with: 'To enable comparison with previous CW-based PAS works, we determined the detection limit of the sensor using...'
4. Line 309: 'Therefore, the sensitivity of DCOS can be pushed to a tenfold increase when the system reaches the shot-noise limit by reducing...' I believe this sentence is referring to the MIMs sensitivity, but it says DCOS. Clarify. Also, for clarity it should be mentioned that this refers to the

shot noise of the probe laser, not the excitation laser. This applies also to line 307 ('away from the shot noise limit for the probe laser.')

5. Line 330: 'We demonstrate the DCOS method, which offers the ppt-level sensitivity.' This is not true, since the ppt sensitivity was not obtained using dual comb excitation, but CW.

6. Line 316: 'Furthermore, accessing the mid-infrared region... will improve the sensitivity, possibly down to the ppt regime.' This statement is correct!

Other comments:

7. Line 54: 'The resonance cavities, however, restrict the wavelength range of operation due to the difficulty of broadband mirror coating.' What is the difficulty? Difficulty to make, purchase? Please be more specific which property of the mirror coatings you are referring to.

8. Line 73: trace gas detection ('gas' is missing)

9. Line 187 and 217: for clarity, I suggest changing 'background' to 'comb envelope'

10. Fig. 3: Please write in the caption that these spectra are recorded with 10% C₂H₂ at atmospheric pressure.

11. Fig. 3c and 4b: Express the y axis in units of cm⁻¹, since a HITRAN fit is shown. This will also help understand how the NNEA was calculated.

12. Line 236 and Supplementary Note 3: The spectrum shown in supplementary Fig. 3 is not Doppler limited. A spectrum is Doppler limited if its linewidth is limited by the Doppler broadening, which is clearly not the case here. The width is 50 MHz (10%) larger than Doppler broadening at room temperature. The pressure broadening for the P(17) line (according to HITRAN) is 0.0733 cm⁻¹/atm (air broadening) and 0.132 cm⁻¹/atm (self-broadening). For pressure of 1000 Pa (0.01 atm) and 50% concentration, this will give $(0.01 * 0.0733 * 0.5 + 0.01 * 0.132 * 0.5) * 30000 = 31$ MHz. So the broadening can be partly explained by collisions, and the line shape should be a Voigt function, not Gaussian. The authors can claim that they measured a low-pressure spectrum, but not that the spectrum is Doppler limited. This needs to be corrected in the manuscript.

13. TABLE 1: the NNEA of this work is not calculated for the spectrum spanning 70 cm⁻¹. It is not even calculated for the spectra from which the 70 cm⁻¹ spectrum was stitched. I understand that the authors can claim that they could stitch 60 spectra spanning 30 GHz each to cover 2 THz. Still, I believe it is misleading to state these numbers next to each other. The authors should state the bandwidth corresponding to the NNEA, because in Refs 16 and 20, also included in this table, the NNEA was evaluated from the spectrum with the spectral width stated in the table.

14. Line 378: State at what conditions α_{\min} was calculated using HITRAN. I assume this is the P(17) line, 1.1 ppb and 1 atm? Here it would also help if the axis of Fig. 4b was expressed in cm⁻¹.

15. Line 384: Allan, not Allen.

16. Supplementary line 171: 'under the atmosphere' should be 'atmospheric pressure'.

Manuscript NCOMMS-23-10200A
“Dual-comb optomechanical spectroscopy”
Reply to Reviewers’ Comments:

We thank the editor and the reviewers for carefully reviewing our manuscript. Reviewer #3 points out that the description and usage of two detection limits (of dual-comb measurement and the sensor itself) need to be further clarified in the manuscript. In the revised version, we have made corresponding modifications (colour highlighting in red) to the reviewers’ concerns. In the following, we provide a point-by-point response (written in blue) to the reviewers’ comments (in black).

Reply to Reviewer #2:

Reviewer #2 (Remarks to the Author):

The authors have addressed my comments from the previous round of review and I would recommend the manuscript for publication.

We are grateful to the Reviewer for supporting our work.

Reply to Reviewer #3:**Reviewer #3 (Remarks to the Author):**

The authors have improved the manuscript following my and the other reviewer's comments. However, the discussion about the sensitivity is still misleading in the paper. In the response letter, the authors clearly distinguish between the detection limit of DCOS and the sensor itself, but in the manuscript this distinction is not clear. This issue must be addressed before the paper is published. I also have a few smaller comments appended at the end of this letter.

We thank the Reviewer for pointing out this issue. We have followed the Reviewer's suggestion and modified the manuscript to clearly distinguish the two detection limits.

In this revised manuscript, we added,

“To enable comparison with cw-based PAS, we determine the detection limit of our sensor by measuring Allan-Werle deviations³⁴⁻³⁶ with a modulated cw laser (Methods). Fig. 4e compares the results with the laser switched on (red) and off (blue). The former shows a minimum measurable concentration of 24 ppt (at 80 s), corresponding to an NNEA of $1.77 \times 10^{-11} \text{ cm}^{-1} \cdot \text{W} \cdot \text{Hz}^{-1/2}$, consistent with the dual-comb result ($1.71 \times 10^{-11} \text{ cm}^{-1} \cdot \text{W} \cdot \text{Hz}^{-1/2}$). The latter indicates the sensor's ultimate detection limit, i.e., 15 ppt for the integration time of 110 s (or $1.3 \times 10^{-11} \text{ cm}^{-1} \cdot \text{W} \cdot \text{Hz}^{-1/2}$). Note that one should distinguish this value (15 ppt for a single element) from the result of DCOS (1.1 ppb for 40 elements).” (p. 15, line 3)

In the response letter, the authors write: ‘There are two concepts of sensitivity in our manuscript; one is the sensitivity measured with the combs (i.e., 1.1 ppb at 100 s for 40 spectral elements), and the other is the detection limit of the optomechanical sensor (i.e., 15 ppt at 110 s based on the intrinsic noise of the sensor). The latter was obtained from a standard Allan-Werle deviation measurement (Fig. 4 e, Laser off) when the excitation laser was switched off’ and ‘We performed the Allan-Werle analysis due to two considerations: 1) to confirm the thermal effect and the detection limit of the MIM sensor (as we have clearly stated in the manuscript); 2) this is a standard process for characterizing a photoacoustic sensor and can be used for direct comparison with other works. In our work, we obtained the dual-comb sensitivity of 1.1 ppb at 100s, which is directly comparable to other dual-comb systems, and we also obtained the detection limit from the Allan-Werle analysis which is directly comparable to other CW-based photoacoustic sensors or methods.’

These descriptions are clear, but they are nowhere to be found in the manuscript. In the manuscript, it is clear that the first detection limit is determined from the SNR of the P(17) line measured at 1 ppm concentration using 150 mW of total comb power distributed over 40 spectral elements (comb modes). Then, the authors measure the noise in the same system filled with N2 using a CW laser with similar total power (200 mW) and without the excitation laser. It is shown that the CW laser does not induce thermal noise (Fig. 4e). However, what is not clear is how the signal measured with the CW laser was converted to concentration. In the response letter, the authors write: ‘The calibration coefficient is obtained using a targeted gas (i.e., C2H2), the details of which we have

described in Methods.’ In the Methods section, line 389, it is stated that ‘For the plots in Fig. 4e, the voltage signal is converted to the equivalent gas concentration, using a measured coefficient of 305.2 $\mu\text{V/ppm}$ for $^{12}\text{C}_2\text{H}_2$.’ It is not explained anywhere HOW this coefficient was measured, while it is the key to the conversion of the measured noise to concentration detection limit. The authors must explain that. I assume that this coefficient is obtained from a calibration measurement of C_2H_2 using the CW laser with 200 mW of power. This coefficient would be different if it was determined using the comb instead (since the power is distributed over the comb lines).

We thank the Reviewer for giving us many wonderful suggestions to improve our manuscript.

Particularly, the reviewer pointed out that “*These descriptions are clear, but they are nowhere to be found in the manuscript.*”

We are sorry that the descriptions in the manuscript were unclear. We have modified the manuscript to clearly distinguish the two detection sensitivities. We added,

“To enable comparison with cw-based PAS, we determine the detection limit of our sensor by measuring Allan-Werle deviations³⁴⁻³⁶ with a modulated cw laser (Methods). Fig. 4e compares the results with the laser switched on (red) and off (blue). The former shows a minimum measurable concentration of 24 ppt (at 80 s), corresponding to an NNEA of $1.77 \times 10^{-11} \text{ cm}^{-1} \cdot \text{W} \cdot \text{Hz}^{-1/2}$, consistent with the dual-comb result ($1.71 \times 10^{-11} \text{ cm}^{-1} \cdot \text{W} \cdot \text{Hz}^{-1/2}$). The latter indicates the sensor’s ultimate detection limit, i.e., 15 ppt for the integration time of 110 s (or $1.3 \times 10^{-11} \text{ cm}^{-1} \cdot \text{W} \cdot \text{Hz}^{-1/2}$). Note that one should distinguish this value (15 ppt for a single element) from the result of DCOS (1.1 ppb for 40 elements).” (p. 15, line 3)

The reviewer then mentioned that “*However, what is not clear is how the signal measured with the CW laser was converted to concentration.*” Also, “*It is not explained anywhere HOW this coefficient was measured, while it is the key to the conversion of the measured noise to concentration detection limit. The authors must explain that. I assume that this coefficient is obtained from a calibration measurement of C_2H_2 using the CW laser with 200 mW of power. This coefficient would be different if it was determined using the comb instead (since the power is distributed over the comb lines).*”

Indeed, the coefficient was obtained from a measurement calibrating the relationship between the output of the lock-in amplifier and the gas concentration (similar to that in refs. 34 and 35). The excitation laser was the cw laser of 200 mW.

We added in the Methods

“For the plots in Fig. 4e, the voltage signal is converted to the equivalent gas concentration, using a coefficient of 305.2 $\mu\text{V/ppm}$ for $^{12}\text{C}_2\text{H}_2$, obtained from a calibration measurement³⁴⁻³⁶ using the 200-mW cw laser.” (p. 22, line 23)

The authors should clearly distinguish between the two detection limits in the paper, they should state that the second, lower, detection limit is for a single element detection only, and that it is

presented to enable comparison with other CW-based PAS methods.

We followed the Reviewer's suggestion and added the following sentences in our revised manuscript.

“To enable comparison with cw-based PAS, we determine the detection limit of our sensor by measuring Allan-Werle deviations³⁴⁻³⁶ with a modulated cw laser (Methods).” (p. 15, line 3)

“Note that one should distinguish this value (15 ppt for a single element) from the result of DCOS (1.1 ppb for 40 elements).” (p. 15, line 9)

Moreover, Reviewer 2 pointed out in their first comment that expressing the detection limit in ppb/ppm units is highly dependent on the selected gas etc. This comment applies also to the detection limit of the sensor itself. The authors write in their response letter: ‘We highlighted the 15-ppt in the abstract because it represented the intrinsic characteristics of our optomechanical sensor and was essential for comparison with other photoacoustic sensors. In contrast, the 1.1-ppb was the sensitivity of our current dual-comb setup, which was affected by, e.g., the excitation comb power and the molecular line strengths.’ This does not make sense, because the 15 ppt limit is also affected by the excitation power and the molecular line strength; it is specific to the particular molecular line chosen. Therefore, the sensor detection limit should also be expressed in units that are independent of the gas, either in the units of absorption coefficient (cm⁻¹), units of NNEA, or in fm/sqrt(Hz).

We agree with the Reviewer that the sensor's detection limit was also influenced by the laser power and the molecular line strength. Following the Reviewer's suggestion, we added the units of NNEA for the sensor's detection limit. The 15 ppt corresponded to an NNEA of $1.3 \times 10^{-11} \text{ cm}^{-1} \cdot \text{W} \cdot \text{Hz}^{-1/2}$. The calculation of NNEA was given in Methods. Because the ppb or ppt units were used in the similar works of dual-comb spectroscopy and PAS, we also kept them in our manuscript for direct comparison with those works (for C₂H₂).

In the revised manuscript (Methods), we wrote,

“The former shows a minimum measurable concentration of 24 ppt (at 80 s), corresponding to an NNEA of $1.77 \times 10^{-11} \text{ cm}^{-1} \cdot \text{W} \cdot \text{Hz}^{-1/2}$, consistent with the dual-comb result ($1.71 \times 10^{-11} \text{ cm}^{-1} \cdot \text{W} \cdot \text{Hz}^{-1/2}$). The latter indicates the sensor's ultimate detection limit, i.e., 15 ppt for the integration time of 110 s (or $1.3 \times 10^{-11} \text{ cm}^{-1} \cdot \text{W} \cdot \text{Hz}^{-1/2}$).” (p. 15, line 6)

In the letter, the authors discuss that the two detection limits are consistent: ‘The sensitivities in the dual-comb and the Allan-Werle measurement (Fig. 4 e) were consistent. In the Allan-Werle measurement, we turned on the CW excitation laser of 200 mW (the maximum power) and obtained a sensitivity limit down to 24 ppt at 100 s (see Fig. 4 e, Laser on), which was slightly above the sensor's detection limit of 15 ppt, indicated by the Laser-off trace (Fig. 4e), because the CW laser induced thermal noise. In the dual-comb measurement, the 1.1 ppb at 100 s was achieved with a total power of 150 mW or 3.75 mW (=150 mW/40) per dual-comb line. Suppose one increases the power per dual-comb line to 200 mW (equal to the CW laser). In that case, the sensitivity will be

improved by 53 (=200mW/3.75mW) times as about 21 ppt at 100 s, provided the photoacoustic signal linearly depends on the excitation power (see Fig. 4d).’ This discussion is correct, but to achieve 200 mW per comb line, while having the same number of comb modes (40), would require 8 W of power in the 30 GHz bandwidth. The authors should include the above discussion in the paper, and comment on what kind of laser they suggest using at these power levels. Will the sensor operate at such high incident power, won’t thermal noise be induced then, etc.?’

We thank the Reviewer for raising this issue. In fact, we wrote the above discussion in the previous response letter to demonstrate that the dual-comb and the Allan-Werle measurements were consistent because our original manuscript confused the Reviewer with the two sensitivity results (i.e., the 1.1 ppb and 15 ppt).

The above discussion is rather straightforward when using the units of NNEA, which the Reviewer also suggested earlier. For instance, the data in Fig. 4e showed the sensitivity limits of 15 ppt (the laser off) and 24 ppt (the laser on), corresponding to NNEAs of 1.3×10^{-11} and $1.77 \times 10^{-11} \text{ cm}^{-1} \cdot \text{W} \cdot \text{Hz}^{-1/2}$, respectively. The latter agreed the dual-comb result ($1.71 \times 10^{-11} \text{ cm}^{-1} \cdot \text{W} \cdot \text{Hz}^{-1/2}$) within 4%.

In the revised manuscript, we added the discussion regarding the sensitivities obtained from the dual-comb and the Allan-Werle measurements to demonstrate their consistency.

“The former shows a minimum measurable concentration of 24 ppt (at 80 s), corresponding to an NNEA of $1.77 \times 10^{-11} \text{ cm}^{-1} \cdot \text{W} \cdot \text{Hz}^{-1/2}$, consistent with the dual-comb result ($1.71 \times 10^{-11} \text{ cm}^{-1} \cdot \text{W} \cdot \text{Hz}^{-1/2}$). The latter indicates the sensor’s ultimate detection limit, i.e., 15 ppt for the integration time of 110 s (or $1.3 \times 10^{-11} \text{ cm}^{-1} \cdot \text{W} \cdot \text{Hz}^{-1/2}$). Note that one should distinguish this value (15 ppt for a single element) from the result of DCOS (1.1 ppb for 40 elements).” (p. 15, line 6)

The Reviewer also pointed out that “*This discussion is correct, but to achieve 200 mW per comb line, while having the same number of comb modes (40), would require 8 W of power in the 30 GHz bandwidth. The authors should include the above discussion in the paper, and comment on what kind of laser they suggest using at these power levels. Will the sensor operate at such high incident power, won’t thermal noise be induced then, etc.?’*”

Technically speaking, erbium-doped fiber amplifiers may boost the comb power to 10 W or more. Customized amplifiers of >10 W are commercially available (e.g., <http://www.microphotons.com/>).

Meanwhile, in the case of a total excitation light power of 8W (or 8~80 nJ for the pulse repetition rate 0.1~1 GHz, beam spot ~0.04 cm²), the MIM (Membrane-In-Mirror) system is expected to remain undamaged due to the high damage threshold of the cavity mirror and membrane constituting it. The LAYERTEC website provides information regarding the damage threshold of the cavity mirror (e.g., 0.1~1 J/cm²; source: LAYERTEC website, link: <https://www.layertec.de/en/capabilities/femtosecond-lasers/high-power/>). Additionally, the paper by Soong, K. presents the damage threshold of the 100-nm-thickness Si₃N₄ membrane (reference: Soong, K. Byer, R. L. & McGuinness, C. *United States: N. p. MOP095*, 277-279 (2011)).

However, we did not recommend using the 8-W excitation power for the dual combs to achieve a sensitivity equivalent to the cw case. Consideration must be given to the photothermal noise and noise introduced by the light source during system operation. Under the 200-mW excitation light, photothermal noise was not observed significantly in our experiment. However, further exploration is necessary to ascertain the system's suitability for high-power applications.

Besides, as mentioned before, we have used the NNEAs to demonstrate the consistency of our measurements. Since the NNEAs were already normalized to the laser power, we did not go into details for commenting the dual-comb powers. Such discussion would lead to a misleading message that we intended to increase the dual-comb power to achieve the ultimate sensitivity (15 or 24 ppt) obtained from the Allan-Werle analysis, which we did not. To avoid such misunderstanding, we did not write the discussion in the previous letter with all the details in the main text.

Instead, we added a discussion in Supplementary Note 1,

“Note that the dual-comb power could be further boosted with additional EDFAs to improve the minimum detectable concentration. However, one should consider the photothermal noise and the intensity noise induced by the high-power amplifier.”

Concerning the Allan-Werle analysis: the authors should cite the original paper where the method was introduced, P. Werle et al, Appl. Phys. B 57, 131–139 (1993). The references that the authors cite now, Ref. 34, Wu et al, and 35, Zhao et al, are also relevant because they show the use of the method in other systems. I note that in both these references the Allan-Werle plot is measured under the same illumination conditions as the sample signal was recorded, but with the sensor filled with N₂. In this work, the authors have changed the illumination conditions from dual-comb to continuous wave.

We thank the Reviewer for reminding us this reference. We added it as

[36] Werle, P., Mücke, R. & Slemr, F. The limits of signal averaging in atmospheric trace-gas monitoring by tunable diode-laser absorption spectroscopy (TDLAS). *Appl. Phys. B* **57**, 131–139 (1993).

The Reviewer pointed out that “*In this work, the authors have changed the illumination conditions from dual-comb to continuous wave.*”

We performed the dual-comb and the Allan-Werle measurements independently. For the latter, we followed the same procedure in Ref. 34 and 35. We used the 200-mW cw laser and C₂H₂ gas for calibration and plotted the Allan-Werle curves with the sensor filled with N₂.

We added in the Methods:

“For the plots in Fig. 4e, the voltage signal is converted to the equivalent gas concentration, using a coefficient of 305.2 μV/ppm for ¹²C₂H₂, obtained from a calibration measurement³⁴⁻³⁶ using the

200-mW cw laser.” (p. 22, line 23)

The phrasings that must be changed in the paper are:

1. Line 29: ‘we measure high-resolution broadband (>2 THz) overtone spectra for acetylene gas and obtain a normalized noise equivalent absorption coefficient of $1.71 \times 10^{-11} \text{ cm}^{-1} \cdot \text{W} \cdot \text{Hz}^{-1/2}$.’ It sounds as if these were obtained simultaneously, while they are not. The NNEA is evaluated in 30 GHz bandwidth. The NNEA would be different when calculated over the 2 THz bandwidth. I suggest saying ‘... and obtain a normalized noise equivalent absorption coefficient of $1.71 \times 10^{-11} \text{ cm}^{-1} \cdot \text{W} \cdot \text{Hz}^{-1/2}$ in a spectrum with 30 GHz simultaneous bandwidth.’

We followed the Reviewer’s suggestion and modified the sentence to “... and obtain a normalized noise equivalent absorption coefficient of $1.71 \times 10^{-11} \text{ cm}^{-1} \cdot \text{W} \cdot \text{Hz}^{-1/2}$ with 30-GHz simultaneous spectral bandwidth.” (p. 2, line 12)

2. Line 31: ‘The optomechanical resonator, with a detection limit down to 15 parts per trillion, allows broadband dual-comb excitation.’ This is misleading as it suggests that the 15 ppt limit is obtained under broadband dual comb excitation. It must be stated that this limit is obtained under CW operation, and it should be expressed in other units.

Since in the abstract we already state a sensitivity of $1.71 \times 10^{-11} \text{ cm}^{-1} \cdot \text{W} \cdot \text{Hz}^{-1/2}$ with 30-GHz simultaneous spectral bandwidth, it is not necessary to give another sensitivity value which is misleading and makes the abstract redundant. Therefore, we delete “with a detection limit down to 15 parts per trillion”.

Now, the sentence reads, “**Importantly**, the optomechanical resonator allows broadband dual-comb excitation.” (p. 2, line 13)

3. Line 270: When starting the discussion about the second detection limit, the authors write: ‘To confirm the detection limit of our system’, which suggests that they continue discussing the same detection limit, which is not the case. Instead, this paragraph should start with: ‘To enable comparison with previous CW-based PAS works, we determined the detection limit of the sensor using...’

We followed the Reviewer’s advice and modified the sentence to “**To enable comparison with cw-based PAS, we determine the detection limit of our sensor by measuring Allan-Werle deviations³⁴⁻³⁶ with a modulated cw laser (Methods).**” (p. 15, line 3)

4. Line 309: ‘Therefore, the sensitivity of DCOS can be pushed to a tenfold increase when the system reaches the shot-noise limit by reducing...’ I believe this sentence is referring to the MIMs sensitivity, but it says DCOS. Clarify. Also, for clarity it should be mentioned that this refers to the shot noise of the probe laser, not the excitation laser. This applies also to line 307 (‘away from the shot noise limit for the probe laser.’)

We followed the Reviewer's advice and modified "DCOS" to "the MIM system".

We also changed the sentence to "... when the system reaches the shot-noise limit (for the probe laser) by reducing..."

And "... which is within 10 dB away from the shot noise limit for the probe laser (see Supplementary Note 6)."

5. Line 330: 'We demonstrate the DCOS method, which offers the ppt-level sensitivity.' This is not true, since the ppt sensitivity was not obtained using dual comb excitation, but CW.

We modified "the ppt-level" to "the ppb-level".

6. Line 316: 'Furthermore, accessing the mid-infrared region... will improve the sensitivity, possibly down to the ppt regime.' This statement is correct!

We sincerely thank the Reviewer for agreeing our statement.

Other comments:

7. Line 54: 'The resonance cavities, however, restrict the wavelength range of operation due to the difficulty of broadband mirror coating.' What is the difficulty? Difficulty to make, purchase? Please be more specific which property of the mirror coatings you are referring to.

It is a technical difficulty to make high-reflection mirrors with broadband coating. We thank the Reviewer for this advice and modified the words to "the technical difficulty of fabricating broadband high-reflection mirrors."

8. Line 73: trace gas detection ('gas' is missing)

We modified "trace detection" to "trace gas detection".

9. Line 187 and 217: for clarity, I suggest changing 'background' to 'comb envelope'

We followed the Reviewer's advice and changed "background" to "comb envelope".

10. Fig. 3: Please write in the caption that these spectra are recorded with 10% C₂H₂ at atmospheric pressure.

Following the Reviewer's suggestion, we added "All the spectra in Fig. 3 are recorded with 10% ¹²C₂H₂ at atmospheric pressure." (Caption of Fig. 3)

11. Fig. 3c and 4b: Express the y axis in units of cm⁻¹, since a HITRAN fit is shown. This will also help understand how the NNEA was calculated.

In the revised manuscript, we updated Fig. 3c and 4b using the units of cm^{-1} for the y axis.

12. Line 236 and Supplementary Note 3: *The spectrum shown in supplementary Fig. 3 is not Doppler limited. A spectrum is Doppler limited if its linewidth is limited by the Doppler broadening, which is clearly not the case here. The width is 50 MHz (10%) larger than Doppler broadening at room temperature. The pressure broadening for the P(17) line (according to HITRAN) is 0.0733 cm-1/atm (air broadening) and 0.132 cm-1/atm (self-broadening). For pressure of 1000 Pa (0.01 atm) and 50% concentration, this will give $(0.01 \cdot 0.0733 \cdot 0.5 + 0.01 \cdot 0.132 \cdot 0.5) \cdot 30000 = 31$ MHz. So the broadening can be partly explained by collisions, and the line shape should be a Voigt function, not Gaussian. The authors can claim that they measured a low-pressure spectrum, but not that the spectrum is Doppler limited. This needs to be corrected in the manuscript.*

We agree with the Reviewer. We removed the “Doppler-limited” from the sentence “We also measure ~~Doppler-limited~~ high-resolution spectra...”

We modified the Supplementary Note 3 and fitted the data with a Voigt function. We wrote,

“We fit the data with a Voigt profile by fixing the Gaussian linewidth at 470 MHz (determined by the Doppler broadening), which gives a Lorentz linewidth of 35 MHz (due to collision broadening).” (Supplementary Note 3)

13. TABLE 1: *the NNEA of this work is not calculated for the spectrum spanning 70 cm-1. It is not even calculated for the spectra from which the 70 cm-1 spectrum was stitched. I understand that the authors can claim that they could stitch 60 spectra spanning 30 GHz each to cover 2 THz. Still, I believe it is misleading to state these numbers next to each other. The authors should state the bandwidth corresponding to the NNEA, because in Refs 16 and 20, also included in this table, the NNEA was evaluated from the spectrum with the spectral width stated in the table.*

We followed the Reviewer’s advice and modified Table 1 using the bandwidth (30 GHz or 1 cm^{-1}) corresponding to the NNEA.

14. Line 378: *State at what conditions α_{min} was calculated using HITRAN. I assume this is the P(17) line, 1.1 ppb and 1 atm? Here it would also help if the axis of Fig. 4b was expressed in cm-1.*

Indeed this calculation was for the data in Fig. 4b which were measured for the P(17) line (1.1 ppb and 1 atm). We added “For Fig. 4b, the minimum detectable absorption coefficient ...” The measurement conditions had been stated in the caption of Fig. 4b.

We also followed the Reviewer’s advice and plot the y axis of Fig. 4b in cm^{-1} .

15. Line 384: *Allan, not Allen.*

We changed “Allen” to “Allan”.

16. *Supplementary line 171: 'under the atmosphere' should be 'atmospheric pressure'.*

We changed “the atmosphere” to “atmospheric pressure”.

Reviewer #3 (Remarks to the Author):

The authors have made the requested changes to the phrasing in the paper.

There are, however, errors in Figures 3(c) and 4(b), where the y axes are now expressed in units of absorption coefficient.

In Fig. 4(b), the absorption coefficient for 1 ppm C₂H₂ at 1 atm should be of the order of 10^{-7} cm⁻¹ (7×10^{-7} cm⁻¹ on resonance). This then gives, for SNR 900, a detection limit of the order of 10^{-10} cm⁻¹, as stated on line 390. However, in Fig. 4(b) the maximum absorption coefficient is 0.4 cm⁻¹, which is 6 orders of magnitude too high. This must be corrected. The authors should also make sure that the minimum detectable absorption coefficient on line 390 (and all NEAS values that are deduced from it) is correct. The ratio between the peak absorption coefficient in Fig. 4(b) and the minimum detectable absorption coefficient on line 390 should be 900, the maximum SNR at 100 s.

Similarly, in Fig. 3(c), for 10% concentration, the absorption coefficient should be of the order of 10^{-2} cm⁻¹, while it is 0.5 cm⁻¹. This must also be corrected.

Manuscript NCOMMS-23-10200B
“Dual-comb optomechanical spectroscopy”
Reply to Reviewers’ Comments:

We thank the editor and the reviewers for reviewing our manuscript. Reviewer #3 points out “*errors in Figures 3(c) and 4(b), where the y axes are now expressed in units of absorption coefficient.*” We appreciate this opportunity to explain the units. In the revised version, we have made corresponding modifications (colour highlighting in red) to the reviewers’ concerns. In the following, we provide a point-by-point response (written in blue) to the reviewers’ comments (in black).

Reply to Reviewer #3:

Reviewer #3 (Remarks to the Author):

The authors have made the requested changes to the phrasing in the paper.

We sincerely thank the Reviewer for accepting our modifications.

There are, however, errors in Figures 3(c) and 4(b), where the y axes are now expressed in units of absorption coefficient. In Fig. 4(b), the absorption coefficient for 1 ppm C₂H₂ at 1 atm should be of the order of 10^{-7} cm⁻¹ (7×10^{-7} cm⁻¹ on resonance). This then gives, for SNR 900, a detection limit of the order of 10^{-10} cm⁻¹, as stated on line 390. However, in Fig. 4(b) the maximum absorption coefficient is 0.4 cm⁻¹, which is 6 orders of magnitude too high. This must be corrected. The authors should also make sure that the minimum detectable absorption coefficient on line 390 (and all NEAS values that are deduced from it) is correct. The ratio between the peak absorption coefficient in Fig. 4(b) and the minimum detectable absorption coefficient on line 390 should be 900, the maximum SNR at 100 s. Similarly, in Fig. 3(c), for 10% concentration, the absorption coefficient should be of the order of 10^{-2} cm⁻¹, while it is 0.5 cm⁻¹. This must also be corrected.

We thank the Reviewer for carefully reviewing our manuscript.

In the previous version, the absorption coefficients (the y axes in Fig. 3(c) and Fig. 4(b)) obtained from the HITRAN database were given for pure C₂H₂. One needs to multiply the y axes by the corresponding concentrations (e.g., 1 ppm or 10^{-6}) we displayed in the figure. This is the reason for “*in Fig. 4(b) the maximum absorption coefficient is 0.4 cm⁻¹, which is 6 orders of magnitude too high.*”

In line 390, we wrote, “For Fig. 4b, the minimum detectable absorption coefficient, α_{\min} , is 4.56×10^{-10} cm⁻¹.” This was achieved for SNR=900 at 100 s. For this value, the peak absorption coefficient was 4.1×10^{-7} cm⁻¹ (i.e., “*The ratio between the peak absorption coefficient in Fig. 4(b) and the minimum detectable absorption coefficient on line 390 should be 900, the maximum SNR at 100 s*”). What we displayed in Fig. 4b was 0.41 cm⁻¹ (without considering the concentration of 1 ppm). Multiplying 0.41 cm⁻¹ with 10^{-6} (i.e., 1 ppm) leads to 4.1×10^{-7} cm⁻¹. Therefore, our

calculations were consistent.

Since our expression was unclear, we followed the Reviewer's suggestions and modified the absorption coefficients for the gas concentration under test. We modified the y axes in Fig. 4(b).

The Reviewer also said *"Similarly, in Fig. 3(c), for 10% concentration, the absorption coefficient should be of the order of 10^{-2} cm^{-1} , while it is 0.5 cm^{-1} . This must also be corrected."*

We also take 10% concentration into consideration and modified Fig. 3(c).

Reviewer #3 (Remarks to the Author):

The axes have been corrected as requested.

Manuscript NCOMMS-23-10200C
“Dual-comb optomechanical spectroscopy”
Reply to Reviewers’ Comments:

We thank the editor and the reviewers for reviewing our manuscript. In the following, we provide a point-by-point response (written in blue) to the reviewers’ comments (in black).

Reply to Reviewer #3:

Reviewer #3 (Remarks to the Author):

The axes have been corrected as requested.

We sincerely thank the Reviewer for accepting our modifications.